# From District to City Scale: The Potential of Water-Sensitive Urban Design (WSUD)

Joachim Schulze *, Simon Gehrmann, Avikal Somvanshi and Annette Rudolph-Cleff

Technical University of Darmstadt, Faculty of Architecture, Urban Design and Development, 64287 Darmstadt, Germany; gehrmann@stadt.tu-darmstadt.de (S.G.); avikals@gmail.com (A.S.); rudolph@stadt.tu-darmstadt.de (A.R.-C.)
* Correspondence: schulze@stadt.tu-darmstadt.de; Tel.: +49-6151-16-22162

**Abstract:** The summer of 2022 was one of the hottest and driest summers that Germany experienced in the 21st century. Water levels in rivers sank dramatically with many dams and reservoirs running dry; as a result, fields could not be irrigated sufficiently, and even power generation and supply were affected. The impact of abnormally high temperatures for extended periods (heatwaves) is not restricted to nature and the economy but is also a considerable public health burden. Experts worldwide agree that these extreme weather events are being driven by climate change and will increase in intensity and frequency in the future. The adverse impact of these extreme weather events multiplies among dense urban environments, e.g., through heat islands. This calls for cities to take action to heat-proof and water-secure their urban developments. Water-Sensitive Urban Design (WSUD) is one such approach to mitigate the aforementioned challenges by leveraging the urban water ecosystem with special attention to the subject of water reclamation, retention, treatment and distribution. This paper introduces and builds upon a prototype of WSUD that centers around an artificial lake as an integrated water resource management system (IWRMS) fed by treated grey water and storm water obtained from two housing blocks flanking the water reservoir. Based on the specifications of this prototype, indicators of site suitability are derived and applied to identify potential locations for replicable projects in the city of Darmstadt. The results confirm the impact WSUD can have: a total of 22 sites with 2527 apartments are found suitable for prototype implementation in Darmstadt. Savings in town water consumption from these 22 sites would add up to 147 million liters. Further benefits include the provision of 24 million liters of irrigation water, storm water retention, adiabatic cooling during heatwave, increased biodiversity and the improvement in livability of the sites and the city.

**Keywords:** water-sensitive urban design; WSUD; integrated water resource management system; IWRMS; climate adaptive cities; climate resilience

## 1. Introduction

Climate change is considered as one of the greatest challenges of the 21st century [1]. Weather conditions in Europe, the United States of America and Asia have increasingly become unusual with both frequency and intensity of heatwaves, droughts and flash floods on the rise [2–4]. These extreme weather conditions have impacted agricultural production, irrigation, energy generation and led to a water supply crisis for cities. This is adversely affecting and even endangering human life. Based on data from the German federal office for statistics during summer months of June to August in 2022, which were characterized by record heat, the number of deaths was significantly higher than the median values from 2018 to 2021 by 9% to 13%. The number of deaths were particularly high in calendar week 29 (18–24 July 2022) at 25%, which was an exceptionally hot week [5]. It is not only extreme heat; even torrential rain and severe flooding are exacting a toll on human life. For instance, the severe flooding catastrophe in the Ahrtal region of Germany in 2021

resulted in devastating damage to houses and infrastructure with a loss of 133 lives [6]. The latest report from The Intergovernmental Panel on Climate Change (IPCC) states that these adverse conditions are not a temporary phenomenon, but the intensity and frequency will increase with "high confidence" [7] (p. 43) in the coming centuries. The 2015 Paris Agreement signed by 196 parties at the 21st UN Climate Change Conference acknowledges that "climate change is a common concern of humankind" [8] (p. 2) and the need for action to limit the increase in global temperature to 1.5 °C. Measures to achieve this objective are directed toward the reduction in greenhouse gas emissions, which are the driving force behind climate change [8] (p. 4). But as recent events show and the IPCC report predicts, climate change is already taking its toll on human life. This calls for tailored ad hoc to medium-term measures to reduce the immediate impact of climate-change-related shocks and stress. The urban settings of cities are more exposed [9] to extremes that culminate in "heat, flooding, water scarcity and droughts" [10] (p. 18). This is due to the dense built environment that characterizes cities and is known to exacerbate the prevailing adverse conditions induced by climate change. The city microclimate, for example, is more prone to heat waves because of the Urban Heat Island (UHI) effect [11]. Studies have also shown that the UHI effect can induce heavy rainfall events [12]. These can subsequently lead to flooding because the sealed (concretized) surfaces lead to heavy runoff, creating a storm water overload that regularly overwhelms the city drainage systems. Droughts also exacerbate the flooding as they damage the urban greenery and dry out the ground, reducing its natural ability to soak in storm water, which adds to heavy runoff when rainfall eventually does set in. In light of these challenges, state agencies have begun to promote climate change mitigation and adaption, e.g., through heat action plans or directives. These include measures referring to the transformation of the built environment such as "conserve and create shaded green spaces and parks, preferably with cooling evaporation areas such as bodies of water [...], reduce heat by creating or keeping clear air channels [...] or reduce degree of soil sealing in open and public squares [...]" [13] (p. 23). Others aim for the protection of vulnerable population groups [13] or water saving incentives [14]. Cities in particular need to take action, a fact which is stressed by studies on urban water scarcity and security [15,16]. After all, an estimated 60% of the world's population will be living in urban environments by 2030 [17] and 80% of Europeans by 2050 [10].

One of the possible adaptation measures that urban settlements can apply is Water-Sensitive Urban Design (WSUD) which incorporates a number of the aspects outlined in the German state heat action plan referenced above [13]. The concept of WSUD originated in the 1990s in Australia [18] and Australian government agencies defined it as "an approach to urban planning and design that aims to integrate the management of the urban water cycle into the urban development process" [19] (p. 1). This paper refers to WSUD as a design principle that, in its essence, reconsiders the management of urban water streams such as storm water (precipitation induced runoff), town water (from tap), greywater (from sinks, showers, bathtubs and laundry) and black water (from toilets or urinals) and intends to integrate these in landscape, building and infrastructure design by means like "reducing potable water demand, minimizing wastewater generation, optimizing the use of water sources" or "promoting a significant degree of water related self-sufficiency" [20] (p. 1). WSUD represents a mitigative measure in the adaptation process of urban environments toward climate change. This is underlined by a report of the European Environment Agency entitled *Urban adaptation in Europe: how cities and towns respond to climate change* [21].

In this paper, a prototype of WSUD is introduced that includes a grey-to-service water treatment, storm water catchment and an artificial lake as a reservoir and central element of landscape design of a residential housing development. Building upon the specifications of this WSUD prototype and investigating the case of the city of Darmstadt, the research in this paper is directed toward the following key questions:

1.  What indicators describe the suitability of a city site for application of a replicable WSUD?
2.  How many potential sites for the application of WSUD exist in the city of Darmstadt?
3.  What impact on the city water cycle of Darmstadt would the application of all WSUD have?
4.  What relevance would this application have in regard to climate change adaptation?

By addressing these questions, this study conducts a multi-scale assessment method based on a district-scale prototype that is multiplied on a city scale. In consideration of the fact that the IWRMS design of the WSUD prototype is one-of-a-kind in Germany and that this study uses real-world as opposed to simulated data, it contributes to filling a gap in research: the quantification of town water savings through district-based WSUD on a city scale.

In the following section on Materials and Methods, the principles of WSUD are outlined, followed by a detailed description of how they are integrated in the WSUD prototype. Next, the city of Darmstadt is introduced. Federal, state and city assessments on climate change are presented in order to evaluate to what extent Darmstadt is already affected by climate change and what climate projections indicate for the future. The last section names the indicators that are derived from the WSUD prototype in order to assess potential WSUD sites. Based on the number of possible WSUD sites in Darmstadt, a grand total of reclaimed and retained water is calculated in order to estimate the overall impact that citywide application could have on the city water cycle.

## 2. Materials and Methods

### 2.1. Water-Sensitive Urban Design

There are multiple definitions for WSUD, but for this paper, it is defined as a design principle that in its essence reconsiders the management of urban water streams such as storm water, town water, grey water and black water and intends to integrate these in landscape, building and infrastructure design. In Germany, WSUD projects referring to the aforementioned definition are rarely found. However, it is worth noting that other design approaches, labeled as "Decentralization" or "Sponge City", can be found in Germany, which are similar to WSUD in objective but differ in scale, design and realization. "Decentralization" or "Sponge City" usually focus on flood protection in the scope of localized storm water management that is mostly designed and realized by engineers and hydrologists with no or little contribution from architects and landscape designers. In the authors' understanding, this is a significant difference and qualifies WSUD as a relatively young new discipline that bridges the gap between urban design and urban water management for building urban climate resilience.

### 2.2. The WSUD Prototype

A WSUD project, which was designed based on an interdisciplinary understanding and the principles of WSUD [22–24], is under construction in the city of Mannheim at the time of the writing of this paper and expected to be finished by 2024. The project contains 74 residential units distributed in two multi-story family houses and offers a variety of different apartment types from 1.5 to 4.5 rooms with a total living space of 5888 m$^2$. The two buildings are arranged in such a way that they form a courtyard with a naturally designed but technically controlled waterbody referred to as the WSUD lake from here onwards (Figure 1). The WSUD lake is a technical water reservoir fed by local storm water and surplus service water reclaimed from greywater from the apartments. It is designed to hold 120 m$^3$ of water on average under normal operating conditions. It also has a provision for extension which enables the holding of an additional 50 m$^3$ of water, which corresponds to a 100-year storm event. In order to allow unlimited access to the lakefront and to prevent any form of fencing, the level of the lake is limited to a depth of 0.4 m. The WSUD lake maintains a steady water level as it receives 100% of the storm water which is collected from the roofs and the façade of the development and the surplus of an

integrated grey-to-service water system of the apartments. Though it is strongly dependent on the season, the storm water inflow accounts for between 60% and 90% of the overall reservoir capacity while service water accounts for the remaining 10% to 40% under normal operation conditions. The grey-to-service water system in the households is designed to collect grey water from showers, sinks and laundry. The grey water is then processed with the help of contemporary water reclamation technology such as ultrafiltration membrane to remove contaminants. This is followed by ultraviolet light exposure (UV) for disinfection, before the majority of it is pumped back to the apartments as service water, while the surplus of service water is spilled in the WSUD lake (Figure 2).

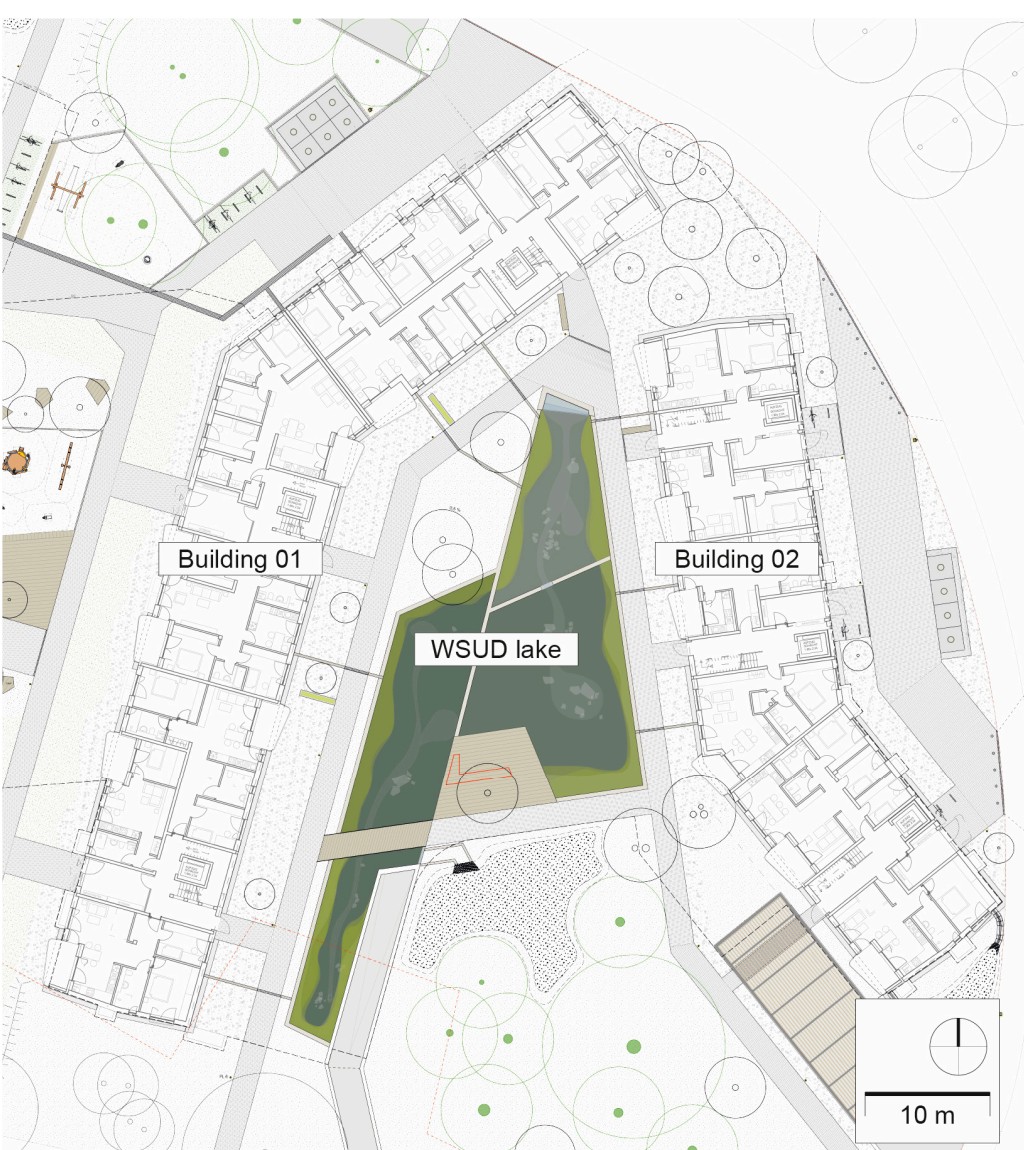

**Figure 1.** Site plan WSUD prototype.

In order to maintain a steady current and to avoid stagnated areas, the WSUD lake is designed in a length to width ratio of 1:7 as a result of specific flow simulations. This also refers to assessments in the field of environmental engineering, in particular, lake management [25]. Furthermore, the lake water is constantly pumped through a sand filter for treatment purposes. The WSUD lake is used for the irrigation of the surrounding green areas between March and October—the extended summer season. The regular capacity is sufficient to meet the irrigation requirements of the development during the normal

dry season, i.e., without any storm water inflow into the WSUD lake. In case of extreme weather events (heat wave or drought) the regular irrigation can be continued until the water level reaches a critical minimum of 25%, after which irrigation needs to be stopped to protect the ecosystem of the WSUD lake. In this situation, the critical minimum of 25% water level of the WSUD lake is maintained by the daily inflow of the surplus of service water which is determined to be sufficient to overcome evaporative losses. In the case of an extreme rain event, the lake extension is activated to hold an additional 50 m$^3$. Water held in the lake extension is designed to be slowly released through infiltration trenches without triggering flooding. This additional water is not retained for the rest of the year for irrigation or reuse purposes.

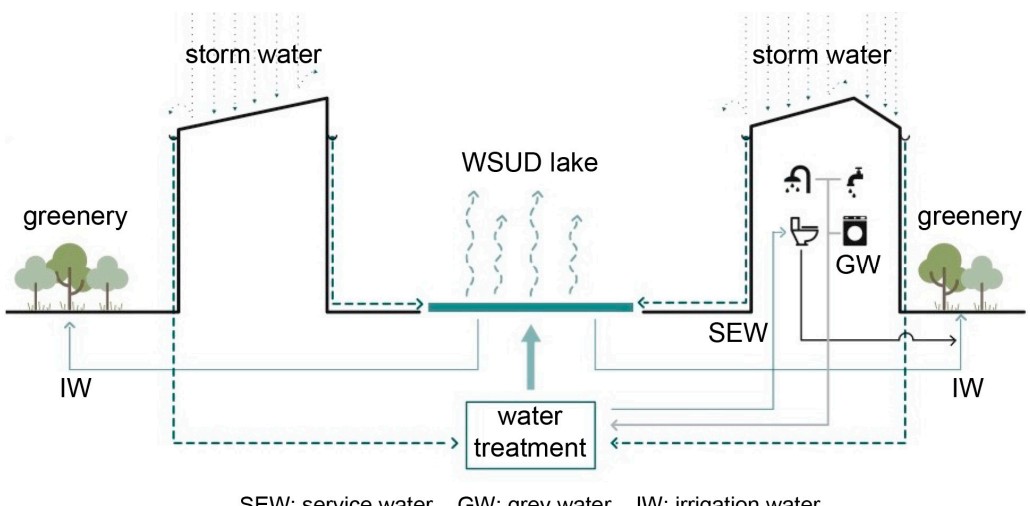

**Figure 2.** Scheme WSUD prototype.

WSUD by closing the water cycle at the decentralized scale can lead to a significant reduction in town water supply. An estimated reduction of up to 40% can be achieved per household. WSUD also eliminates the burden of irrigating common greens within the development with town water, which is estimated to further lower town water demand from the site by 20%. The estimated payback period for the initial investment in WSUD is about 25 years based on the savings in town water consumption and wastewater fee payments. While the decentralized water management technology itself is not new [26–30], the WSUD prototype stands out by combining treated grey water, i.e., service water and storm water, in the WSUD lake. This approach distinguishes it from other notable WSUD projects in Germany such as the "Hamburg WATER CYCLE" [31] or the "BUGA Heilbronn" [32]. In addition, the way the technology is applied under WSUD and connects to people via design and architecture is unique in this project. WSUD also provides an improvement in quality of life and a reduction in vulnerability to extreme weather, which is not easily captured in the aforementioned technical and financial details. It is critical to acknowledge the importance of design and architecture in unlocking the full potential of technical intervention like decentralized water management. Design can help leverage it to increase biodiversity by creating a constantly green oasis while providing the community with a highly resilient structure which can mitigate the effects of climate change.

### 2.3. The City of Darmstadt

The city of Darmstadt is a mid-sized university town located in the federal state of Hesse 30 km south of Frankfurt am Main. Darmstadt has a population of 163,435 and an area of 122 km$^2$ [33], making it the fourth biggest city within the federal state of Hesse. The city is an important center of education, research and science. Darmstadt has major employers in the information and communication, healthcare and manufacturing sectors [34]. Darmstadt is part of the Frankfurt Rhine-Main metropolitan region which

includes the cities of Aschaffenburg, Offenbach and Frankfurt am Main. Frankfurt Rhine-Main is one of the eleven metropolitan regions in Germany and is home to a population of 5.8 million people and the location of numerous influential political institutions and corporate headquarters [35]. Climate change adaptation has gained traction within the municipality of Darmstadt with multiple strategies being developed and implemented in the last decade. In 2013, an integrated concept for climate protection was set up [36] which was followed by a 25-point immediate action plan for climate protection in 2020. In 2021, the city of Darmstadt established the office for climate protection and climate adaptation, which meanwhile has grown to a staff of over ten employees.

Exposure to Climate Change

In 2021, the German Environment Agency (UBA) published its report, *The Risks of Climate in Germany,* where it clearly confirms that Germany will be affected by climate change nationwide [37,38]. It also noted that adverse effects such as heatwaves or heavy storm water will not be distributed evenly but impact differs regionally [38]. The overview map in Figure 3 illustrates how different parts of Germany, e.g., the coasts, the northwest and the mountainous regions, are exposed to either a minor, strong or very strong increase in four categories of risk, i.e., average temperature, heat, heavy storm water and drought.

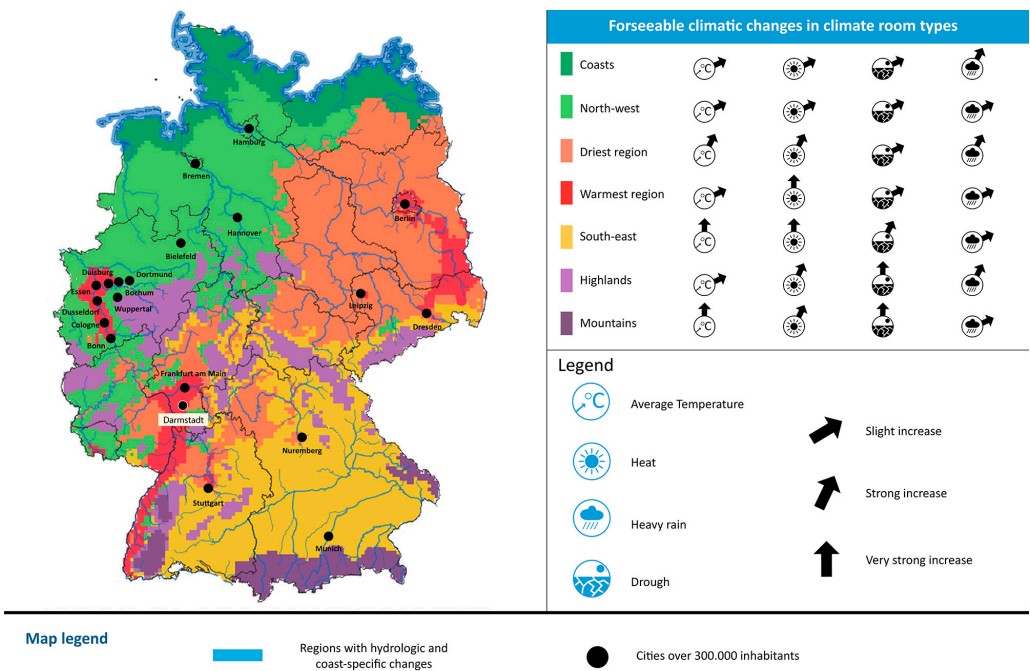

**Figure 3.** Regional impact of climate change in Germany.

Climate projections for the city of Darmstadt predict a very strong increase in heat risk. While a minor increase in the remaining three risk categories is also predicted. Figure 3 also shows that Darmstadt is located in the warmest region of Germany. In addition, the report states that densely populated urban areas such as the "Ruhrgebiet" or the "Rhein-Main-Neckar Region" will be affected significantly more by weather extremes [38].

The findings of UBA have been seconded by climate change studies performed at the state and city levels. The 2017 integrated climate protection plan of the state of Hesse that incorporates climate projections carried out by the Potsdam Institute for Climate Impact Research confirms that the degree and characteristics of climate-change-related effects vary regionally, and the south of the state of Hesse including the region around the city of Darmstadt will be especially affected by high temperatures and long-lasting heat [39]. Studies on climate change carried out or commissioned by the municipality of Darmstadt complement the federal and the state reports by providing data of higher granularity on a

district scale. The semi-annual report on statistics in 2020, for example, includes a special issue on climate change in Darmstadt [40]. This report makes use of data provided by a number of temperature metering stations positioned throughout the city. From 1961 to 2019, the annual mean temperature has risen from 10 °C to 11 °C with summer and winter seasons experiencing the greatest increase in monthly average temperatures. The number of summer days with temperatures exceeding 25 °C has increased from an average of 43 days between 1961 and 1990 to 76 days in the time span from 2015 to 2019. The number of hot days characterized by a temperature that exceeds 35 °C has tripled while average frequency of heat waves has increased eight-fold for the same reference time frames. A heat waves is defined as "a multi-day period with extraordinary thermal stress" [40] (p. 11). The climate reports paint a slightly different concern regarding storm water risk to the city of Darmstadt. The number of rainy days per year has not changed significantly, but the mean annual storm water has decreased. From 2015 to 2019 average storm water per year amounted to 639 mm, which is 13% less than from 1961 to 1990. Furthermore, the distribution of storm water has shifted. Spring and summer are getting dryer, whereas the winter season is receiving more storm water. From 2015 to 2019, storm water volume dropped by more than 20 mm in the months of July and August, which is significant. Nevertheless, the Hessian Agency for Nature Conservation, Environment and Geology labels the city of Darmstadt as "increasingly at risk" [40] (p. 18) to heavy storm water events. A 2016 study commissioned by the city of Darmstadt also provides further details on the city microclimate and its exposure to climate-change-related effects [41]. All climate risk and vulnerability studies are unanimous that the city of Darmstadt is already suffering ill-effects of climate change, which are predicted to worsen in the coming future. The risk to the city is predominantly posed by increase in temperature and decrease in storm water during the summer. The city already has a tough baseline, given its location in the hottest region of Germany, and shows pronounced UHI effects in its inner core. The predicted further increases in heat and decreases in storm water will lead to numerous problems in the city. One of the most obvious will be drought, as high heat leads to water evaporation that in turn aggravates drought [40]. Thus, dryness will also pose a major problem in the future.

*2.4. Further Data and Sources*

To map and assess potential WSUD sites in Darmstadt following data were acquired:

1. For the assessment of potential WSUD sites, a leading municipal housing association provides data on their real estate located in Darmstadt. Altogether, its building stock includes 17,000 units, which is comparable to the size of the housing association realizing the WSUD prototype in Mannheim that maintains 15,000 units. This is an important indicator for transferability from the setting of Mannheim to Darmstadt. And both operate in mid-size cities. The data provided include address, number of buildings per site, number of building levels, number of apartments, total living space, date of erection and status of refurbishment.

2. For the assessment and the mapping of potential WSUD sites, the spatial data of the city of Darmstadt including infrastructure, plots, buildings, topographic and aerial photos are provided by the land surveyors office of the city of Darmstadt.

3. Data on monthly, daily and hourly storm water for Darmstadt are retrieved online from the German Weather Service (DWD) [42,43] and assessed for a ten-year period.

## 3. Results

### *3.1. Indicators for WSUD Application*

The WSUD prototype presented in the previous section serves as a template for the assessment of potential sites in Darmstadt that are suited for the citywide application of a replicable WSUD. To ensure citywide coverage, the paper is focused on identifying existing building sites with suitability for WSUD application rather than green field developments. The main intention for focusing on the existing building stock is simply that the ratio of existing buildings is much higher than those which are being newly built in Darmstadt. Thus, emphasizing existing sites will, in all probability, lead to the identification of more WSUD applications than focusing on sites that are currently in development. This approach to maximize impact is supported by studies carried out in the context of the German energy transition, which have concluded that the biggest reduction in energy consumption can be achieved by the energy-focused refurbishment of the existing building stock [44]. The energy-focused refurbishment of the building stock in Germany is an ongoing process that is driven by state and federal subsidization as well as legal requirements with the long-term goal of achieving climate neutrality by 2050 [45]. From an economic, strategic and logistic point of view, it will be reasonable to combine WSUD application with energy-focused refurbishment measures, because there is an overlap in the overall climate objectives of both programs. Moreover, "resource efficiency" in a holistic sense [46] (p. 15) as defined by the Sustainable Development Goals (SDG) to "make cities and human settlements inclusive, safe, resilient and sustainable" [46] (p. 15) includes both water and energy. Hence, non-refurbished sites awaiting energy-focused renovation are particularly suited for WSUD application.

Indicators (financial and O and M viability) for site section derived from the prototype specifications are (Table 1):

**Table 1.** Indicators for WSUD site assessment.

| Reference to Prototype | |
| --- | --- |
| **Indicator** | **Specification** |
| Building use | Residential |
| Ownership | Housing association |
| Operation | Housing association |
| Number of units and/or grand total of living space | $\geq$70 and or $\geq$5900 m$^2$ |
| **Reference to existing site application** | |
| State of refurbishment (one of the three specifications must apply) | Building stock that was erected at least 40 years ago and that has never been renovated |
| | Building stock that was renovated at least 40 years ago |
| | Building stock that by the year 2030 will have been unrenovated for 40 years |
| Available open space | Must be sufficient to position the WSUD lake |

1. Building use, ownership and operation

These indicators refer to the usage of the buildings in the prototype which are 100% residential and the building cluster being owned and operated by a city housing association. The building stock assessed in this study comply with both indicators.

2. Number of units and/or grand total of living space

The WSUD prototype is connected to a total of 74 apartment units located in the two buildings flanking the courtyard. Altogether, these provide 5888 m$^2$ of living space.

Both figures are indicators of financial feasibility. This is a necessity for the housing association executing the prototype, otherwise the project would not have been approved by the executive board. With this number of units, or rather living space, involved, the payback for additional costs of the WSUD prototype is estimated to be within 25 years from the savings in town water use for household and irrigation purposes and by a reduction in wastewater fee payments.

Indicators (logistical viability) for site selection derived from existing site surveys are outline below (Table 1):

1.    State of refurbishment

WSUD application in existing building sites would require extensive civil work which includes landscaping, concrete work and the installation of the water treatment unit that additionally requires separate piping in the buildings for grey water and service water [47]. In order to mount this system, the installation shafts in the buildings need to be opened. This is an expensive undertaking, but costs can be rationalized if they are aligned with other refurbishment measures or incorporated within the scope of a full renovation. This can be achieved by clubbing WSUD applications with federal- and state-mandated and subsidized energy-focused refurbishments. Therefore, to ensure the city-scale implementation of WSUD, it is deemed important to link site selection with the refurbishment status of the potential sites. But the timespan when a building is due for renovation can only be defined as an approximation because the lifespan of building materials depends on numerous factors [48] that can lead to disparities between the expected and the actual lifespan, and even though a building might be in a state of neglect, it may not mean that the building operator will take immediate action. In this paper, a 40-year timespan is defined as the age for refurbishment based on the average lifespan of different piping materials [49] when they have reached a state of "major damage that cannot be repaired due to technical or financial reasons" [49] (p. 12). Based on the aforementioned rationale, indicators of state of refurbishment are assessed in three ways: (1) building stock that was erected at least 40 years ago and that has never been renovated, (2) building stock that was renovated at least 40 years ago and (3) building stock that by the year 2030 will have been unrenovated for 40 years.

2.    Available open space

The site must have sufficient open space for the WSUD lake to be positioned.

*3.2. Shortlisted Sites for WSUD Application*

A total of 496 sites were examined in Darmstadt (Figure 4) against the indicators as described in Table 1. In the initial assessment, 19 sites met all the indicators. In the follow-up assessment, three additional sites were shortlisted despite not meeting all the stated indicators. These three sites, namely #3, #7 and #9, fell short of the minimum threshold indicator "number of units and/or grand total of living space" by only less than 9%, which the authors feel is not a significant enough shortfall to disregard their potential for inclusion in the study. Finally, a grand total of 22 sites were shortlisted for assessing the city-scale WSUD application. These sites account for 4.4% of all sites examined in the city of Darmstadt. Since most of the shortlisted sites are significantly large housing developments, they together represent 19% of all apartment units and 20% of total living space examined in the city with a population of about 6300 tenants.

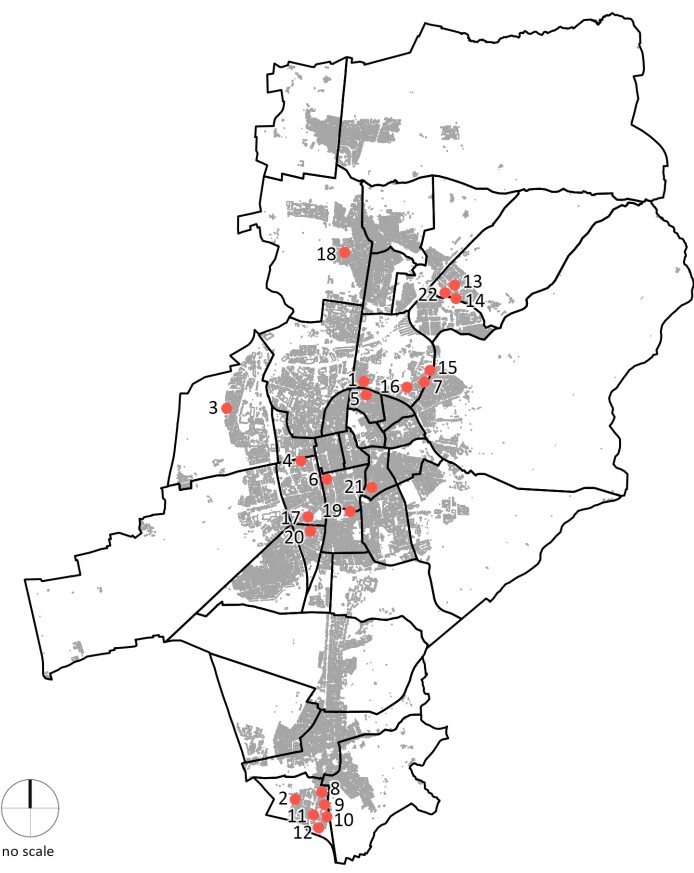

**Figure 4.** Distribution of 22 sites in Darmstadt.

*3.3. WSUD Lake Dimensioning and Water Balance*

In order to assess the overall impact of citywide WSUD prototype implementation, a WSUD lake is dimensioned for each of the 22 shortlisted sites. Based on the dimensions and the water balance of each WSUD lake, a grand total of reused service water, gained irrigation water and retained storm water has been calculated. The lake design and dimensioning calculations refer to specifications and average values applied in the prototype, such as averages of town water consumption per capita, averages of the amount of grey water and black water per capita or averages of the number of tenants for each unit. The lake balancing method refers to standards used in hydrology studies [50]. Climatology data on storm water in Darmstadt from 2011 to 2021 has been sourced from DWD. Data on urban morphologies (Table A1) are used to calculate unsealed space, i.e., the greenery, based on average ratios. Further values include standard irrigation rates (Table A2). Water inflow includes storm water and the surplus of service water from the households; water outtake refers to evaporation and the water provided for irrigation purposes. It must be noted that water inflow by runoff from greenery and water outtake in the form of infiltration from the lake bank and bottom are not taken into consideration because the prototype design specifications do not do so. Furthermore, calculations in this study are performed based on monthly water inflow and outtake in the WSUD lake in order to achieve a monthly water balance. This is the appropriate granularity to meet the objectives of this paper, but the application of approximations, standard values and monthly as opposed to daily or hourly balancing lead to limitations. The precise calculation of roofs and greenery and the application of exact figures on town water consumption or irrigation demands would refine the results. The conduction of hourly simulations would lead to an even better understanding of the inflow and outflow of water and could consider the holiday season with reduced water consumption per capita, which results in less service water inflow. This could further improve the lake design. The calculations consist of two preliminary steps

leading to the main calculation for the monthly lake water balance. For conciseness, this section focuses on the description of the main calculation. The preliminary steps are only presented in brief but are outlined in Appendix A of this paper. Table 2 lists all variables and parameters included in the dimensioning and balancing of the WSUD lakes.

**Table 2.** WSUD lake variables and parameters.

| Description | | Additional Notes |
|---|---|---|
| **Variables** | | |
| Total number of units [-] | $U$ | - |
| $\Sigma$ base area of buildings [m$^2$] | $A_{base}$ | Proxy for calculation of SW inflow |
| $\Sigma$ open space of site [m$^2$] | $A_{open\ space}$ | Includes sealed (concretized) and unsealed area |
| $\Sigma$ unsealed open space of site (greenery) [m$^2$] | $A_{greenery}$ | Is calculated based on average ratios |
| $\Sigma$ irrigated open space of site [m$^2$] | $A_{irrigation}$ | Is calculated by subtraction of lake area |
| $\Sigma$ litres of irrigation water (IW) [l] | $(\Sigma_{liters\ IW\ pre})$ | Total liters IW from pre-dimensioning |
| $\Sigma$ litres of irrigation water (IW) [l] | $(\Sigma_{liters\ IW\ lake})$ | Total liters IW referring to lake |
| $\Sigma$ litres of irrigation water (IW) [l] | $(\Sigma_{liters\ IW})$ | Total liters IW |
| $\Sigma$ greywater [L/per day] | $GW$ | Total GW based on number of units |
| $\Sigma$ reused greywater as service water [L/per day] | $SEW_{reuse}$ | Total GW that is reused |
| $\Sigma$ surplus service water [L/per day] | $SEW_{surplus}$ | Is GW minus $SEW_{reuse}$ |
| Area lake pre-dimension | $L_{areapre}$ | Pre-dimension of main water body |
| Area lake retention | $L_{arearetention}$ | Area of lake extension for SW retention |
| Area lake | $L_{area}$ | Area of main water body |
| Width of lake [m] | $L_{width}$ | - |
| Length of lake [m] | $L_{length}$ | - |
| **Parameters** | | |
| Depth of main water body [m] | $L_{depth\ main\ water\ body}$ | Is 0.,4 m according to the prototype |
| Depth of lake extension for SW retention [m] | $L_{depth\ retention}$ | Is 0.2 m |
| Ratio lake width to lake length [-] | $L_{ratio}$ | Is 1:7 according to the prototype |
| $\varnothing \Sigma$ greywater [L/per resident and day] | $\varnothing GW$ | Average value |
| $\varnothing \Sigma$ reused GW as SEW [L/per resident and day] | $\varnothing SEW_{reuse}$ | Average value |
| $\varnothing \Sigma$ surplus SEW [L/per resident and day] | $\varnothing SEW_{surplus}$ | Is $\varnothing GW$ minus $\varnothing SEW_{reuse}$ |
| $\varnothing \Sigma$ surplus SEW [L/per resident and mth] | $\varnothing SEW_{surplusmth}$ | Monthly SEW surplus |
| Ratios unsealed to sealed open space [%] | $R_{open\ space}$ | Are taken from a study on urban morphologies |
| Duration of irrigation [-] | $IR_{duration}$ | Is defined as 15 days |
| $\varnothing$ daily irrigation demand [mm] | $\varnothing IR_{day}$ | Average values |
| $\varnothing$ monthly irrigation demand [mm] | $\varnothing IR_{mth}$ | Average values |
| Daily storm water maximum [mm] | $SW_{max}$ | Based on SW data from 2011–2021 |
| $\varnothing$ monthly storm water [mm] | $\varnothing SW_{mth}$ | Based on SW data from 2011–2021 |
| $\varnothing$ monthly evaporation [mm] | $\varnothing EV_{mth}$ | Average values |

### 3.3.1. Preliminary Steps

Preliminary steps define the size of the main water body of the WSUD lake that is to be used for irrigation purposes and an extension area intended to temporarily retain storm water from extreme rainfall events. The irrigation capacity refers to the amount of water

that is necessary to fully irrigate the standard greenery of each site for a period of 15 days. Retention, on the other hand, is based on the amount of storm water that would add to the lake level in the case of a 100-year heavy storm water event. The calculations incorporate average values on urban typologies [51] (Table A1) and irrigation rates [52] (Table A2). Retention capacity refers to the maximum daily storm water event for Darmstadt (Table A3). The dimensions of the lakes are restricted to a depth of 0.4 m and a width to length ratio of 1:7, as outlined in the description of the prototype.

### 3.3.2. Monthly Water Balance

The preliminary steps are the basis for the monthly lake water balance (Figure 5). Water inflow through storm water refers to monthly averages (Table 3). Because site data do not include information on the nature of the roofs, $SW_{mth}$ is multiplied with the base area ($A_{base}$) of the buildings to approximate the total of storm water inflow. Additionally, inflow includes the surplus of service water per month ($SEW_{surplus}$) based on average consumption values applied in the design of the prototype and multiplied with the number of units (U). Lake outtake includes irrigation water and evaporation. The German weather service (DWD) provides data on evaporation for a lake depth of two meters. The deviation in the depth of the prototype is negligible because lake depth is not taken into account for the calculation of evaporation [53]. The total amount of irrigation water withdrawn from the lake is calculated by multiplying monthly irrigation averages ($ØIR_{mth}$) with the irrigation area ($A_{irrigation}$) of the site.

$$A_{base} \times ØSW_{mth} = \sum \text{litres from SW inflow} \tag{1}$$

$$U \times ØSEW_{surplusmth} = \sum \text{litres from SEW surplus} \tag{2}$$

$$A_{irrigation} \times ØIR_{mth} = \sum \text{litres of irrigation} \tag{3}$$

$$L_{area} \times ØEV_{mth} = \sum \text{litres of evaporation} \tag{4}$$

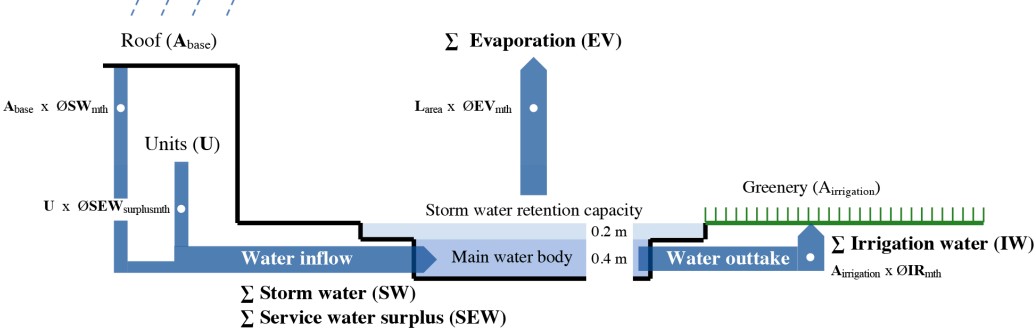

**Figure 5.** Scheme monthly lake water balance.

**Table 3.** Monthly storm water average Darmstadt.

| Month | Ø Σ SW [MM] per Month 2011–2021 | Month | Ø Σ SW [MM] per Month 2011–2021 |
|---|---|---|---|
| January | 62 | July | 59 |
| February | 41 | August | 66 |
| March | 39 | September | 49 |
| April | 38 | October | 52 |
| May | 68 | November | 50 |
| June | 62 | December | 68 |

As an example, Figure 6 shows the lake water balance sheets for sites 03 and 04 alongside the corresponding site plan. Irrigation rates are reduced in gradations of $\frac{3}{4}$, $\frac{1}{2}$, $\frac{1}{3}$, $\frac{1}{4}$ and $\frac{1}{8}$ referring to the greenery that is watered. As stated in the description of the prototype, lake levels must not fall below a capacity of 25%. Apart from this requirement lake levels must reach 100% when the irrigation phase begins in April.

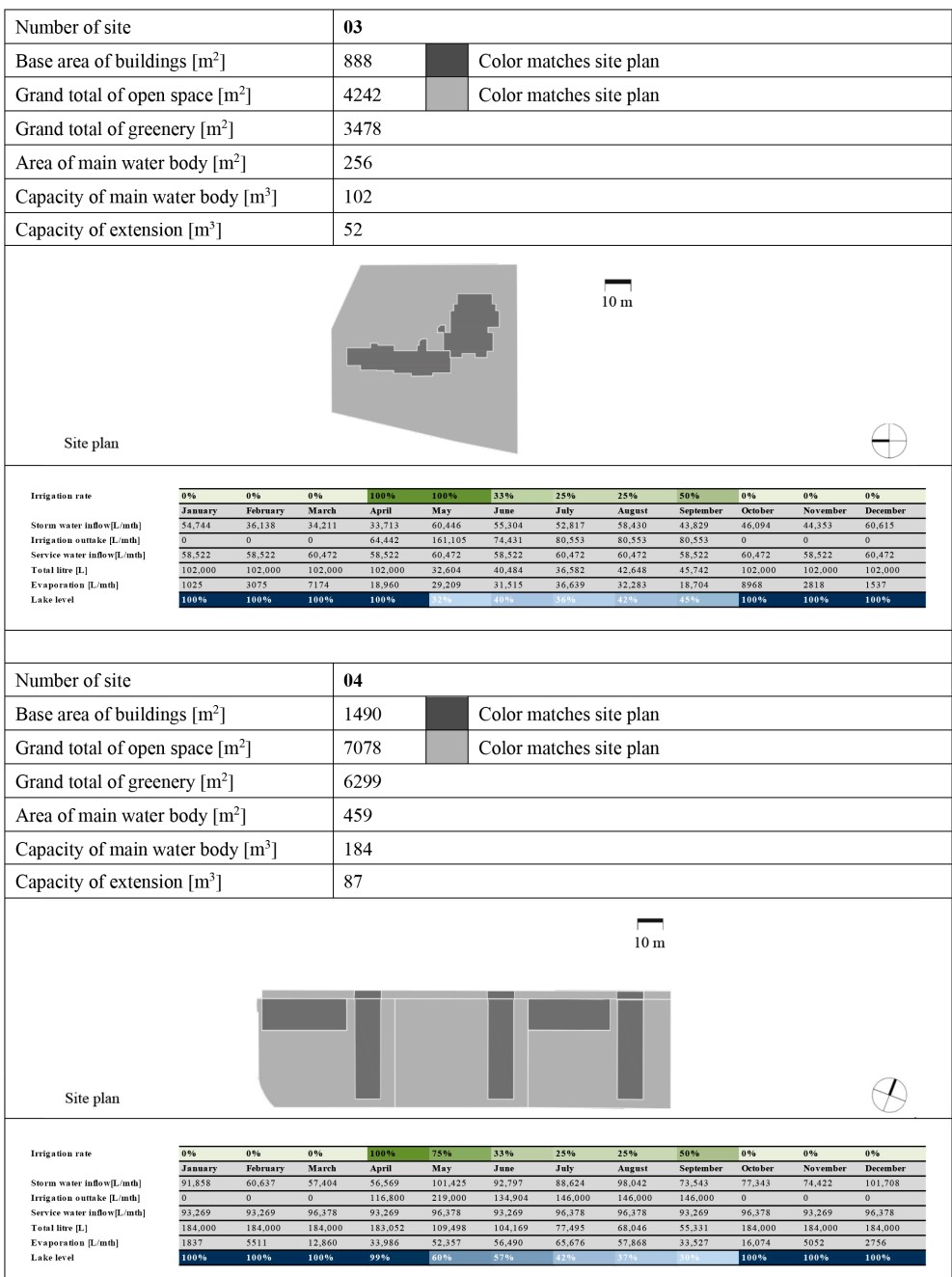

**Figure 6.** Example site sheet 03 and 04.

The remaining 21 site sheets (Figures A1–A10) can be found in the Appendix B of this paper alongside data on monthly storm water in Darmstadt (Table A4) and site assessment (Table A5).

### 3.4. Impact on the City Water Cycle

Based on the final calculations, altogether (a) 123 million liters of town water could be saved by WSUD prototype application per year from the 22 sites. This refers to the reuse of service water in toilets and washing machines in the households. Irrigation water provided by the lake would account for (b) 24 million liters per year. Two million liters is the total storm water retention capacity. Thus, the total savings in town water add up to 147 million liters (a + b) per year, which corresponds to the consumption of 3200 inhabitants. Under normal circumstances, this water would need to be retrieved from groundwater. However, it must be noted that despite all the designing and engineering permutations and combinations, only four sites can maintain full irrigation throughout the entire year with WSUD application. These four sites have a relatively higher ratio of built-up area to open space ranging between 0.35 and 0.51. This means they have much higher storm water runoff and less greenery to irrigate compared to other sites. In these cases, the supply of storm water and surplus of service water exceed the irrigation demands from April to September. The remaining 18 sites must lower irrigation levels between May and September ranging from 25% to 75%, otherwise the lake level would fall below the critical 25%. This is due to their ratio of built-up area to open space being low, and they range between 0.09 and 0.21. The reduced amount of irrigation water can either be distributed evenly or can be allocated to priority irrigation spots.

### 3.5. Further Limitations and Obstacles

Apart from the indicators listed in Table 1 applied for WSUD site assessment, there are a few caveats that need to be named and considered. These do not affect the application but may have implications on the WSUD lake design. These caveats would need further investigation and consideration in case a WSUD project is ultimately executed on site. The first caveat refers to the area that needs to be provided for the fire brigade so it can access the site in case of a fire. Length, width and position of this area are based on set requirements but to a limited extent are negotiated with the local fire brigade and tailored according to the site [54]. Nevertheless, this area cannot overlap with the WSUD lake; therefore, lake design has to be adjusted to accommodate this. The second caveat relates to the existing trees on a site. It can be a challenge to preserve old and full-grown trees while positioning the WSUD lake on the site. It is important to point out that trees are beneficial for climate resilience in an urban context [55] and should be preserved; therefore, WSUD lake design has to be adjusted to accommodate such trees. As noted earlier, splitting the lake into more than one basin is possible and could help address both aforementioned caveats.

Other noteworthy obstacles were witnessed in the planning process of the WSUD prototype. Generally speaking, storm water management and in particular the infiltration into the soil has to be approved by the environmental agency. The management of grey and service water, on the other hand, has to be approved by the health agency. Since the IWRMS combines storm, grey and service water, there was no clear responsibility. Only a special permit signed by both units enabled the continuation of planning and realization. In addition, the housing association in Mannheim insisted on a proof-of-concept ensuring that lake levels do not fall below the critical minimum in summertime. These examples demonstrate two challenges: it was new for the city administration to handle a project that did not comply with water management standards, and on the client side, there was recurring insecurity in the prototype's flawless operation.

## 4. Conclusions and Discussion

Climate change is already impacting weather patterns across the world, and the scientific consensus is that it will only get worse. Urban environments, due to their inherent form, are aggravating the impact of climate change on their residents, increasing the urgency to implement adaptation measures. Assessments carried out for Germany and the city of Darmstadt in particular underline this necessity. The WSUD prototype presented in this paper could be one of numerous interconnected measures taken in order to mitigate

the impact of climate change. The assessment carried out in this paper provides evidence that WSUD can help the city during drought by reducing dependency on town water for local irrigation and household use. It also helps provide thermal relief during heatwave due to the adiabatic cooling effect of the WSUD lake on the development sites. During extreme storm water events, the WSUD can soften the pressure on the city storm water drainage system by locally managing the extra storm water and reducing the possibility of flooding. Apart from these climate adaptation benefits, the WSUD also improves quality of life. It enhances landscape design, it helps to preserve biodiversity and it provides adiabatic cooling.

The other question that this paper attempted to explore was that of scalability and application to an existing city. The paper finds through its investigation of the city of Darmstadt that a considerable number of housing developments already have the logistical suitability—physical space and appropriate vintage warranting renovation—for implementing WSUD. The paper found that 22 WSUD projects can be implemented in the city serving an impressive 2527 apartment units covering 155,153 m² of living space in the city of Darmstadt. Annual savings of about 147 million liters of town water and local storm water retainment capacity of 2 million liters are achieved in this assessment. These results are despite the scope of this assessment being limited to sites owned and serviced by one party—municipal housing associations—which limited the evaluation to 496 sites. Additionally, the paper adopted very strict indicators that further limited selection to existing sites that have not been renovated for at least 40 years.

Another critical finding of the paper is that even with WSUD application it would not be possible to maintain 100% irrigation through WSUD of standard greeneries during the driest portion of the summer. This is because standard irrigation rates applied in this paper [51] lead to rather high irrigation outtakes, as they are based on water-intensive greenery. This means that to obtain 100% irrigation for the whole summer, there needs to be additional tweaking in landscape design by moving away from the standard water-intensive greenery toward native drought-tolerant plants and trees. Further, the viability of the greenery can be supported by increasing the saturation capacity of the soil. These two options were not investigated in this paper as they were beyond the study scope but could be explored as part of future research. Other research opportunities include the quantification of the added benefits of adiabatic cooling, the improvement of local biodiversity and overall quality of life from this exercise, the investigation of other potential sites for WSUD prototype application incorporating mixed forms of ownership and the refinement of WSUD prototype lake design through simulations. Additionally, investigating other cities could provide further insight on the question of scalability. In this case, transferability relies first and foremost on matching the indicators described in Chapter 3.1. City size, on the other hand, can vary.

The benefits and the water saving potential of WSUD outlined in this study suggest a change in city water politics in Germany. As mentioned earlier, administrations need to develop procedures that can handle new innovative approaches like the WSUD prototype that do not correspond to the well-established understanding of water management. In addition, WSUD should be adopted as a climate adaptation policy mandate and receive support with subsidies as performed with energy conservation renovations. But this calls for a more frequent consideration of WSUD by planners which, in contrast to known design principles, also emphasizes improving the quality of life. Both could pave the way for further WSUD projects in German cities. In conclusion, this paper demonstrates that WSUD has clear potential to help build a city's climate resilience and reduce the pressure on town water systems. But it must also be noted that this paper is based on a specific WSUD prototype application, and it is only one element in the transformation process toward climate resilience. Other measures need to be investigated.

**Author Contributions:** Conceptualization, J.S.; methodology, J.S. and S.G.; investigation, J.S. and S.G.; data curation, J.S.; writing—original draft preparation, J.S. and S.G.; writing—review and editing, A.S. and A.R.-C.; visualization, J.S. and S.G. All authors have read and agreed to the published version of the manuscript.

**Funding:** This research received no external funding.

**Data Availability Statement:** Data are found in the manuscript and Appendices A and B.

**Acknowledgments:** This work has been funded by the LOEWE initiative (Hesse, Germany) within the emergenCITY center.

**Conflicts of Interest:** The authors declare no conflicts of interest.

**Appendix A**

$$\mathbf{A}_{\text{open space}} \times \mathbf{R}_{\text{open space}} = \mathbf{A}_{\text{greenery}} \tag{A1}$$

$$\mathbf{A}_{\text{greenery}} \times \mathbf{ØIR}_{\text{day}} \times 15 = \sum \text{liters IW pre} \tag{A2}$$

$$\mathbf{L}_{\text{width}} \times \mathbf{L}_{\text{length}} \times 0.4 = \mathbf{L}_{\text{areapre}} \tag{A3}$$

(with a ratio of 1:7 and in accordance with (A2))

$$\mathbf{A}_{\text{base}} \times \mathbf{SW}_{\text{max}} = \sum \text{litres from SW inflow} \tag{A4}$$

$$\mathbf{L}_{\text{width}} \times \mathbf{L}_{\text{length}} \times 0.2 = \mathbf{L}_{\text{areaextension}} \tag{A5}$$

(with a ratio of 1:7 and in accordance with (A4))

**Table A1.** Ratios of greenery to open space of common urban morphologies.

| Type | Detached Houses | Row Houses | Row Structures with Low Density | Row Structures with High Density | Block Structure | Rural Houses | Historical Center | City Center |
|---|---|---|---|---|---|---|---|---|
| $R_{\text{open space}}$ | 0.9 | 0.97 | 0.89 | 0.82 | 0.54 | 0.62 | 0.57 | 100 |

**Table A2.** Average irrigation demands.

|  | J | F | M | A | M | J | J | A | S | O | N | D |
|---|---|---|---|---|---|---|---|---|---|---|---|---|
| mm per mth | 0 | 0 | 0 | 20 | 50 | 70 | 100 | 100 | 50 | 0 | 0 | 0 |
| mm per day | 0 | 0 | 0 | 0.67 | 1.61 | 2.33 | 3.23 | 3.23 | 1.67 | 0 | 0 | 0 |
| Ø mm per day | 2.12 (from April to September) | | | | | | | | | | | |

$$\mathbf{L}_{\text{areapre}} \times \mathbf{ØIR}_{\text{day}} \times 15 = \sum \text{liters IW lake} \tag{A6}$$

$$\sum \text{liters IW pre} - \sum \text{litres IW lake} = \sum \text{liters IW} \tag{A7}$$

$$\mathbf{L}_{\text{width}} \times \mathbf{L}_{\text{length}} \times 0.4 = \mathbf{L}_{\text{area}} \tag{A8}$$

(with a ratio of 1:7 and in accordance with (A7))

$$\mathbf{A}_{\text{greenery}} - \mathbf{L}_{\text{area}} = \mathbf{A}_{\text{irrigation}} \tag{A9}$$

**Table A3.** Maximum daily storm water event Darmstadt.

| Day | Hour | SW [mm] | Day | Hour | SW [mm] |
|---|---|---|---|---|---|
| 26 August 2011 | 00:00 | 0 | 26 August 2011 | 12:00 | 0 |
| 26 August 2011 | 01:00 | 0 | 26 August 2011 | 13:00 | 0 |
| 26 August 2011 | 02:00 | 21.4 | 26 August 2011 | 14:00 | 0 |
| 26 August 2011 | 03:00 | 22.8 | 26 August 2011 | 15:00 | 0 |
| 26 August 2011 | 04:00 | 0 | 26 August 2011 | 16:00 | 0 |
| 26 August 2011 | 05:00 | 0 | 26 August 2011 | 17:00 | 0 |
| 26 August 2011 | 06:00 | 0 | 26 August 2011 | 18:00 | 0 |
| 26 August 2011 | 07:00 | 0 | 26 August 2011 | 19:00 | 1.1 |
| 26 August 2011 | 08:00 | 0 | 26 August 2011 | 20:00 | 10.4 |
| 26 August 2011 | 09:00 | 0 | 26 August 2011 | 21:00 | 1.6 |
| 26 August 2011 | 10:00 | 0 | 26 August 2011 | 22:00 | 0.6 |
| 26 August 2011 | 11:00 | 0 | 26 August 2011 | 23:00 | 0.6 |
| **Sum total: 58.5** | | | | | |

## Appendix B

**Table A4.** Monthly storm water Darmstadt.

| Year | Month | Σ SW [mm] | Year | Month | Σ SW [mm] |
|---|---|---|---|---|---|
| 2011 | January | 59.2 | 2017 | January | 25.5 |
| 2011 | February | 18.4 | 2017 | February | 27.2 |
| 2011 | March | 16.1 | 2017 | March | 48.6 |
| 2011 | April | 12.3 | 2017 | April | 14.9 |
| 2011 | May | 15.3 | 2017 | May | 80.8 |
| 2011 | June | 66.9 | 2017 | June | 52.6 |
| 2011 | July | 75.8 | 2017 | July | 130.3 |
| 2011 | August | 151 | 2017 | August | 93.6 |
| 2011 | September | 42.1 | 2017 | September | 61.9 |
| 2011 | October | 28.2 | 2017 | October | 50.1 |
| 2011 | November | 1.2 | 2017 | November | 108.6 |
| 2011 | December | 104.6 | 2017 | December | 74.9 |
| 2012 | January | 70.6 | 2018 | January | 79.3 |
| 2012 | February | 10.5 | 2018 | February | 15.8 |
| 2012 | March | 19.3 | 2018 | March | 55.8 |
| 2012 | April | 42.5 | 2018 | April | 54.4 |
| 2012 | May | 48.5 | 2018 | May | 43.5 |
| 2012 | June | 89.6 | 2018 | June | 22.6 |
| 2012 | July | 123.9 | 2018 | July | 8 |
| 2012 | August | 62.7 | 2018 | August | 15.9 |

**Table A4.** *Cont.*

| Year | Month | Σ SW [mm] | Year | Month | Σ SW [mm] |
|------|-------|-----------|------|-------|-----------|
| 2012 | September | 42.3 | 2018 | September | 34.6 |
| 2012 | October | 60.5 | 2018 | October | 7.9 |
| 2012 | November | 45.2 | 2018 | November | 24.7 |
| 2012 | December | 94.1 | 2018 | December | 113 |
| 2013 | January | 47.3 | 2019 | January | 58.7 |
| 2013 | February | 47 | 2019 | February | 10.2 |
| 2013 | March | 29 | 2019 | March | 50.8 |
| 2013 | April | 78.1 | 2019 | April | 43.1 |
| 2013 | May | 136.3 | 2019 | May | 109.2 |
| 2013 | June | 69.4 | 2019 | June | 49 |
| 2013 | July | 26.8 | 2019 | July | 41.2 |
| 2013 | August | 72.1 | 2019 | August | 49.6 |
| 2013 | September | 83.2 | 2019 | September | 75.6 |
| 2013 | October | 90.3 | 2019 | October | 93.4 |
| 2013 | November | 68.6 | 2019 | November | 63.8 |
| 2013 | December | 42 | 2019 | December | 69 |
| 2015 | January | 88.6 | 2020 | January | 37.4 |
| 2015 | February | 25 | 2020 | February | 108.8 |
| 2015 | March | 31.2 | 2020 | March | 66 |
| 2015 | April | 25.2 | 2020 | April | 14.7 |
| 2015 | May | 10.1 | 2020 | May | 67.1 |
| 2015 | June | 24 | 2020 | June | 48.5 |
| 2015 | July | 28.2 | 2020 | July | 21.9 |
| 2015 | August | 40.4 | 2020 | August | 75.5 |
| 2015 | September | 62.1 | 2020 | September | 38.4 |
| 2015 | October | 16.8 | 2020 | October | 65.1 |
| 2015 | November | 74.7 | 2020 | November | 22.2 |
| 2015 | December | 30.5 | 2020 | December | 91.5 |
| 2016 | January | 72.4 | 2021 | January | 77.4 |
| 2016 | February | 99.8 | 2021 | February | 44.2 |
| 2016 | March | 45.6 | 2021 | March | 22.8 |
| 2016 | April | 72.2 | 2021 | April | 22.2 |
| 2016 | May | 81.4 | 2021 | May | 88.4 |
| 2016 | June | 103.1 | 2021 | June | 97 |
| 2016 | July | 43.5 | 2021 | July | 95.1 |
| 2016 | August | 35.9 | 2021 | August | 61.2 |
| 2016 | September | 27.4 | 2021 | September | 25.9 |
| 2016 | October | 53 | 2021 | October | 53.7 |
| 2016 | November | 48.4 | 2021 | November | 42 |
| 2016 | December | 9.8 | 2021 | December | 53.1 |

**Table A5.** Shortlisted sites.

| Indicator | Value | Number |
|---|---|---|
| Number of units and or grand total of living space | $\geq$70 and or $\geq$5900 m$^2$ | 1 |
| State of refurbishment | Building stock that was erected at least 40 years ago and that has never been renovated | 2 |
| | Building stock that was renovated at least 40 years ago | 3 |
| | Building stock that by the year 2030 will have been unrenovated for 40 years | 4 |

| Number of site [-] | Levels per building [-] | Erection year | State of refurbishment | Number of units [-] | Living space [m$^2$] | Indicator match |
|---|---|---|---|---|---|---|
| 1 | 4 | 1929 | Not refurbished | 10 | 505 | |
| 1 | 4 | 1929 | Not refurbished | 10 | 505 | |
| 1 | 4 | 1929 | Not refurbished | 10 | 517 | |
| 1 | 4 | 1929 | Not refurbished | 9 | 502 | |
| 1 | 4 | 1929 | Not refurbished | 10 | 502 | |
| 1 | 4 | 1929 | Not refurbished | 10 | 522 | |
| 1 | 4 | 1929 | Not refurbished | 10 | 520 | 1 and 2 |
| 1 | 4 | 1929 | Not refurbished | 9 | 507 | |
| 1 | 4 | 1929 | Not refurbished | 9 | 505 | |
| 1 | 4 | 1929 | Not refurbished | 10 | 517 | |
| 1 | 4 | 1929 | Not refurbished | 10 | 511 | |
| 1 | 4 | 1929 | Not refurbished | 10 | 521 | |
| Total | | | | 117 | 6135 | |
| 2 | 2 | 1950 | Not refurbished | 6 | 304 | |
| 2 | 2 | 1950 | Not refurbished | 6 | 300 | |
| 2 | 2 | 1950 | Not refurbished | 6 | 301 | |
| 2 | 2 | 1950 | Not refurbished | 6 | 323 | |
| 2 | 2 | 1950 | Not refurbished | 6 | 323 | |
| 2 | 2 | 1950 | Not refurbished | 6 | 323 | |
| 2 | 2 | 1950 | Not refurbished | 6 | 301 | |
| 2 | 2 | 1950 | Not refurbished | 6 | 304 | |
| 2 | 2 | 1950 | Not refurbished | 6 | 300 | |
| 2 | 2 | 1950 | Not refurbished | 6 | 301 | 1 and 2 |
| 2 | 2 | 1950 | Not refurbished | 6 | 301 | |
| 2 | 2 | 1950 | Not refurbished | 5 | 264 | |
| 2 | 2 | 1950 | Not refurbished | 6 | 309 | |
| 2 | 2 | 1950 | Not refurbished | 6 | 307 | |
| 2 | 2 | 1950 | Not refurbished | 6 | 323 | |
| 2 | 2 | 1950 | Not refurbished | 6 | 323 | |
| 2 | 2 | 1950 | Not refurbished | 6 | 316 | |
| 2 | 2 | 1950 | Not refurbished | 6 | 322 | |
| 2 | 2 | 1950 | Not refurbished | 6 | 382 | |

**Table A5.** *Cont.*

| Indicator | | | Value | | | | Number |
|---|---|---|---|---|---|---|---|
| 2 | 2 | 1950 | Not refurbished | 6 | 322 | | |
| 2 | 2 | 1950 | Not refurbished | 6 | 322 | | |
| 2 | 2 | 1950 | Not refurbished | 6 | 323 | | |
| 2 | 2 | 1950 | Not refurbished | 6 | 303 | | |
| 2 | 2 | 1950 | Not refurbished | 6 | 303 | | |
| 2 | 2 | 1950 | Not refurbished | 6 | 343 | | |
| 2 | 2 | 1950 | Not refurbished | 5 | 291 | | |
| Total | | | | 154 | 8139 | | |
| 3 | 9 | 1971 | Not refurbished | 38 | 3048 | | |
| 3 | 9 | 1971 | Not refurbished | 16 | 1360 | | |
| 3 | 9 | 1971 | Not refurbished | 10 | 608 | | 1 and 2 |
| Total | | | | 64 | 5015 | | |
| 4 | 5 | 1951 | Not refurbished | 9 | 507 | | |
| 4 | 5 | 1951 | Not refurbished | 10 | 619 | | |
| 4 | 5 | 1951 | Not refurbished | 15 | 721 | | |
| 4 | 5 | 1951 | Not refurbished | 9 | 507 | | |
| 4 | 5 | 1951 | Not refurbished | 10 | 619 | | 1 and 2 |
| 4 | 5 | 1951 | Not refurbished | 15 | 721 | | |
| 4 | 5 | 1951 | Not refurbished | 9 | 507 | | |
| 4 | 5 | 1951 | Not refurbished | 10 | 619 | | |
| 4 | 5 | 1951 | Not refurbished | 15 | 721 | | |
| Total | | | | 102 | 5542 | | |
| 5 | 4 | 1930 | Not refurbished | 10 | 394 | | |
| 5 | 4 | 1930 | Not refurbished | 10 | 397 | | |
| 5 | 4 | 1930 | Not refurbished | 9 | 374 | | |
| 5 | 4 | 1930 | Not refurbished | 9 | 585 | | |
| 5 | 4 | 1930 | Not refurbished | 10 | 533 | | |
| 5 | 4 | 1930 | Not refurbished | 9 | 408 | | |
| 5 | 4 | 1930 | Not refurbished | 9 | 460 | | |
| 5 | 4 | 1930 | Not refurbished | 5 | 280 | | |
| 5 | 4 | 1930 | Not refurbished | 4 | 271 | | 1 and 2 |
| 5 | 4 | 1930 | Not refurbished | 10 | 384 | | |
| 5 | 4 | 1930 | Not refurbished | 8 | 315 | | |
| 5 | 4 | 1930 | Not refurbished | 10 | 396 | | |
| 5 | 4 | 1930 | Not refurbished | 8 | 312 | | |
| 5 | 4 | 1930 | Not refurbished | 10 | 391 | | |
| 5 | 4 | 1930 | Not refurbished | 9 | 396 | | |
| 5 | 4 | 1930 | Not refurbished | 4 | 281 | | |
| 5 | 4 | 1930 | Not refurbished | 10 | 415 | | |
| Total | | | | 144 | 6591 | | |

**Table A5.** *Cont.*

| Indicator | | | Value | | | Number |
|---|---|---|---|---|---|---|
| 6 | 5 | 1984 | Not refurbished | 11 | 837 | |
| 6 | 5 | 1984 | Not refurbished | 10 | 692 | |
| 6 | 5 | 1984 | Not refurbished | 20 | 1198 | 1 and 4 |
| 6 | 5 | 1984 | Not refurbished | 20 | 1255 | |
| 6 | 5 | 1984 | Not refurbished | 12 | 938 | |
| Total | | | | 73 | 4920 | |
| 7 | 8 | 1969 | Not refurbished | 19 | 1965 | |
| 7 | 8 | 1969 | Not refurbished | 25 | 1872 | 1 and 2 |
| 7 | 8 | 1969 | Not refurbished | 18 | 1904 | |
| Total | | | | 62 | 5741 | |
| 8 | 3 | 1966 | Not refurbished | 6 | 414 | |
| 8 | 3 | 1966 | Not refurbished | 6 | 433 | |
| 8 | 3 | 1966 | Not refurbished | 6 | 433 | |
| 8 | 3 | 1966 | Not refurbished | 6 | 414 | |
| 8 | 3 | 1966 | Not refurbished | 6 | 433 | |
| 8 | 3 | 1966 | Not refurbished | 6 | 414 | 1 and 2 |
| 8 | 12 | 1966 | Not refurbished | 60 | 3566 | |
| 8 | 4 | 1966 | Not refurbished | 8 | 573 | |
| 8 | 4 | 1966 | Not refurbished | 8 | 673 | |
| 8 | 4 | 1966 | Not refurbished | 8 | 618 | |
| 8 | 4 | 1966 | Not refurbished | 8 | 618 | |
| Total | | | | 128 | 8586 | |
| 9 | 14 | 1967 | Not refurbished | 69 | 5161 | 1 and 2 |
| Total | | | | 69 | 5161 | |
| 10 | 12 | 1968 | Not refurbished | 72 | 4293 | 1 and 2 |
| Total | | | | 72 | 4293 | |
| 11 | 11 | 1974 | Not refurbished | 46 | 3277 | 1 and 2 |
| 11 | 11 | 1974 | Not refurbished | 61 | 4677 | |
| Total | | | | 107 | 7954 | |
| 12 | 9 | 1975 | Not refurbished | 30 | 2296 | |
| 12 | 5 | 1975 | Not refurbished | 13 | 764 | |
| 12 | 9 | 1975 | Not refurbished | 14 | 1049 | |
| 12 | 9 | 1975 | Not refurbished | 46 | 3401 | |
| 12 | 5 | 1975 | Not refurbished | 13 | 764 | |
| 12 | 9 | 1975 | Not refurbished | 46 | 3289 | 1 and 2 |
| 12 | 5 | 1975 | Not refurbished | 13 | 764 | |
| 12 | 9 | 1975 | Not refurbished | 14 | 1049 | |
| 12 | 9 | 1975 | Not refurbished | 45 | 3214 | |
| 12 | 9 | 1975 | Not refurbished | 45 | 3238 | |
| 12 | 5 | 1975 | Not refurbished | 13 | 764 | |

**Table A5.** *Cont.*

| Indicator | | | Value | | | Number |
|---|---|---|---|---|---|---|
| 12 | 9 | 1975 | Not refurbished | 16 | 1198 | |
| 12 | 9 | 1975 | Not refurbished | 51 | 3819 | |
| 12 | 5 | 1975 | Not refurbished | 16 | 954 | |
| 12 | 9 | 1975 | Not refurbished | 27 | 2066 | |
| 12 | 9 | 1975 | Not refurbished | 45 | 3214 | |
| 12 | 5 | 1975 | Not refurbished | 13 | 764 | |
| 12 | 9 | 1975 | Not refurbished | 14 | 1049 | |
| Total | | | | 474 | 33,655 | |
| 13 | 3 | 1982 | Not refurbished | 12 | 773 | |
| 13 | 3 | 1982 | Not refurbished | 10 | 817 | |
| 13 | 3 | 1982 | Not refurbished | 8 | 608 | |
| 13 | 3 | 1982 | Not refurbished | 10 | 817 | |
| 13 | 3 | 1982 | Not refurbished | 8 | 607 | |
| 13 | 3 | 1982 | Not refurbished | 12 | 772 | |
| 13 | 3 | 1982 | Not refurbished | 11 | 691 | 1 and 4 |
| 13 | 3 | 1982 | Not refurbished | 7 | 473 | |
| 13 | 3 | 1982 | Not refurbished | 11 | 693 | |
| 13 | 3 | 1982 | Not refurbished | 11 | 691 | |
| 13 | 3 | 1982 | Not refurbished | 12 | 714 | |
| 13 | 3 | 1982 | Not refurbished | 7 | 535 | |
| Total | | | | 119 | 8190 | |
| 14 | 3 | 1983 | Not refurbished | 7 | 418 | |
| 14 | 3 | 1983 | Not refurbished | 8 | 463 | |
| 14 | 1 | 1983 | Not refurbished | 1 | 90 | |
| 14 | 1 | 1983 | Not refurbished | 1 | 90 | |
| 14 | 3 | 1983 | Not refurbished | 8 | 463 | |
| 14 | 1 | 1983 | Not refurbished | 1 | 90 | |
| 14 | 1 | 1983 | Not refurbished | 1 | 90 | 1 and 4 |
| 14 | 3 | 1983 | Not refurbished | 8 | 607 | |
| 14 | 3 | 1983 | Not refurbished | 8 | 696 | |
| 14 | 3 | 1983 | Not refurbished | 10 | 817 | |
| 14 | 3 | 1983 | Not refurbished | 11 | 693 | |
| 14 | 3 | 1983 | Not refurbished | 6 | 464 | |
| 14 | 3 | 1983 | Not refurbished | 8 | 584 | |
| Total | | | | 78 | 5562 | |
| 15 | 3 | 1984 | Not refurbished | 60 | 2929 | |
| 15 | 1 | 1984 | Not refurbished | 35 | 1738 | 1 and 4 |
| Total | | | | 95 | 4667 | |

**Table A5.** *Cont.*

| Indicator | | | Value | | | Number |
|---|---|---|---|---|---|---|
| 16 | 4 | 1990 | Not refurbished | 21 | 1209 | |
| 16 | 4 | 1990 | Not refurbished | 29 | 1660 | |
| 16 | 4 | 1990 | Not refurbished | 29 | 1660 | 1 and 4 |
| Total | | | | 79 | 4530 | |
| 17 | 3 | 1986 | Not refurbished | 8 | 573 | |
| 17 | 3 | 1986 | Not refurbished | 2 | 131 | |
| 17 | 3 | 1986 | Not refurbished | 2 | 131 | |
| 17 | 3 | 1986 | Not refurbished | 4 | 304 | |
| 17 | 3 | 1986 | Not refurbished | 8 | 601 | |
| 17 | 3 | 1986 | Not refurbished | 2 | 118 | |
| 17 | 3 | 1986 | Not refurbished | 2 | 144 | |
| 17 | 3 | 1986 | Not refurbished | 2 | 144 | |
| 17 | 3 | 1986 | Not refurbished | 2 | 118 | |
| 17 | 3 | 1986 | Not refurbished | 8 | 601 | |
| 17 | 3 | 1986 | Not refurbished | 4 | 304 | 1 and 4 |
| 17 | 3 | 1986 | Not refurbished | 2 | 131 | |
| 17 | 3 | 1986 | Not refurbished | 2 | 131 | |
| 17 | 3 | 1986 | Not refurbished | 8 | 573 | |
| 17 | 3 | 1986 | Not refurbished | 2 | 144 | |
| 17 | 3 | 1986 | Not refurbished | 2 | 144 | |
| 17 | 3 | 1986 | Not refurbished | 2 | 118 | |
| 17 | 3 | 1986 | Not refurbished | 8 | 601 | |
| 17 | 3 | 1986 | Not refurbished | 8 | 601 | |
| 17 | 3 | 1986 | Not refurbished | 2 | 118 | |
| Total | | | | 80 | 5731 | |
| 18 | 3 | 1989 | Not refurbished | 33 | 1576 | |
| 18 | 3 | 1989 | Not refurbished | 27 | 1197 | |
| 18 | 3 | 1989 | Not refurbished | 35 | 1839 | 1 and 4 |
| Total | | | | 95 | 4612 | |
| 19 | 3 | 1960 | Not refurbished | 47 | 1395 | |
| 19 | 4 | 1960 | Not refurbished | 23 | 848 | |
| 19 | 3 | 1960 | Not refurbished | 38 | 799 | 1 and 2 |
| 19 | 3 | 1960 | Not refurbished | 24 | 1067 | |
| Total | | | | 132 | 4109 | |
| 20 | 4 | 1926 | Not refurbished | 8 | 386 | |
| 20 | 4 | 1926 | Not refurbished | 8 | 385 | |
| 20 | 4 | 1926 | Not refurbished | 8 | 386 | |
| 20 | 3 | 1926 | Not refurbished | 8 | 387 | 1 and 2 |
| 20 | 4 | 1926 | Not refurbished | 8 | 387 | |
| 20 | 4 | 1926 | Not refurbished | 8 | 387 | |

**Table A5.** *Cont.*

| Indicator | | | Value | | | Number |
|---|---|---|---|---|---|---|
| 20 | 3 | 1926 | Not refurbished | 8 | 387 | |
| 20 | 3 | 1926 | Not refurbished | 8 | 387 | |
| 20 | 3 | 1926 | Not refurbished | 8 | 385 | |
| 20 | 3 | 1926 | Not refurbished | 8 | 372 | |
| 20 | 3 | 1926 | Not refurbished | 8 | 400 | |
| 20 | 3 | 1926 | Not refurbished | 8 | 383 | |
| Total | | | | 96 | 4632 | |
| 21 | 1 | 1951 | Not refurbished | 6 | 366 | |
| 21 | 1 | 1951 | Not refurbished | 6 | 345 | |
| 21 | 1 | 1951 | Not refurbished | 3 | 148 | |
| 21 | 1 | 1951 | Not refurbished | 6 | 368 | |
| 21 | 1 | 1951 | Not refurbished | 6 | 345 | |
| 21 | 1 | 1951 | Not refurbished | 8 | 452 | |
| 21 | 1 | 1951 | Not refurbished | 6 | 343 | |
| 21 | 1 | 1951 | Not refurbished | 8 | 451 | |
| 21 | 1 | 1951 | Not refurbished | 6 | 343 | 1 and 2 |
| 21 | 1 | 1951 | Not refurbished | 8 | 450 | |
| 21 | 1 | 1951 | Not refurbished | 6 | 345 | |
| 21 | 1 | 1951 | Not refurbished | 8 | 451 | |
| 21 | 1 | 1951 | Not refurbished | 8 | 445 | |
| 21 | 1 | 1951 | Not refurbished | 10 | 322 | |
| 21 | 1 | 1951 | Not refurbished | 8 | 217 | |
| Total | | | | 103 | 5390 | |
| 22 | 4 | 1983 | Not refurbished | 8 | 575 | |
| 22 | 4 | 1983 | Not refurbished | 8 | 585 | |
| 22 | 4 | 1983 | Not refurbished | 8 | 585 | |
| 22 | 4 | 1983 | Not refurbished | 8 | 585 | |
| 22 | 4 | 1983 | Not refurbished | 8 | 585 | |
| 22 | 4 | 1983 | Not refurbished | 8 | 508 | |
| 22 | 3 | 1983 | Not refurbished | 6 | 438 | 1 and 4 |
| 22 | 3 | 1983 | Not refurbished | 6 | 438 | |
| 22 | 3 | 1983 | Not refurbished | 6 | 438 | |
| 22 | 3 | 1983 | Not refurbished | 6 | 407 | |
| 22 | 3 | 1983 | Not refurbished | 6 | 415 | |
| 22 | 3 | 1983 | Not refurbished | 6 | 438 | |
| Total | | | | 84 | 5996 | |

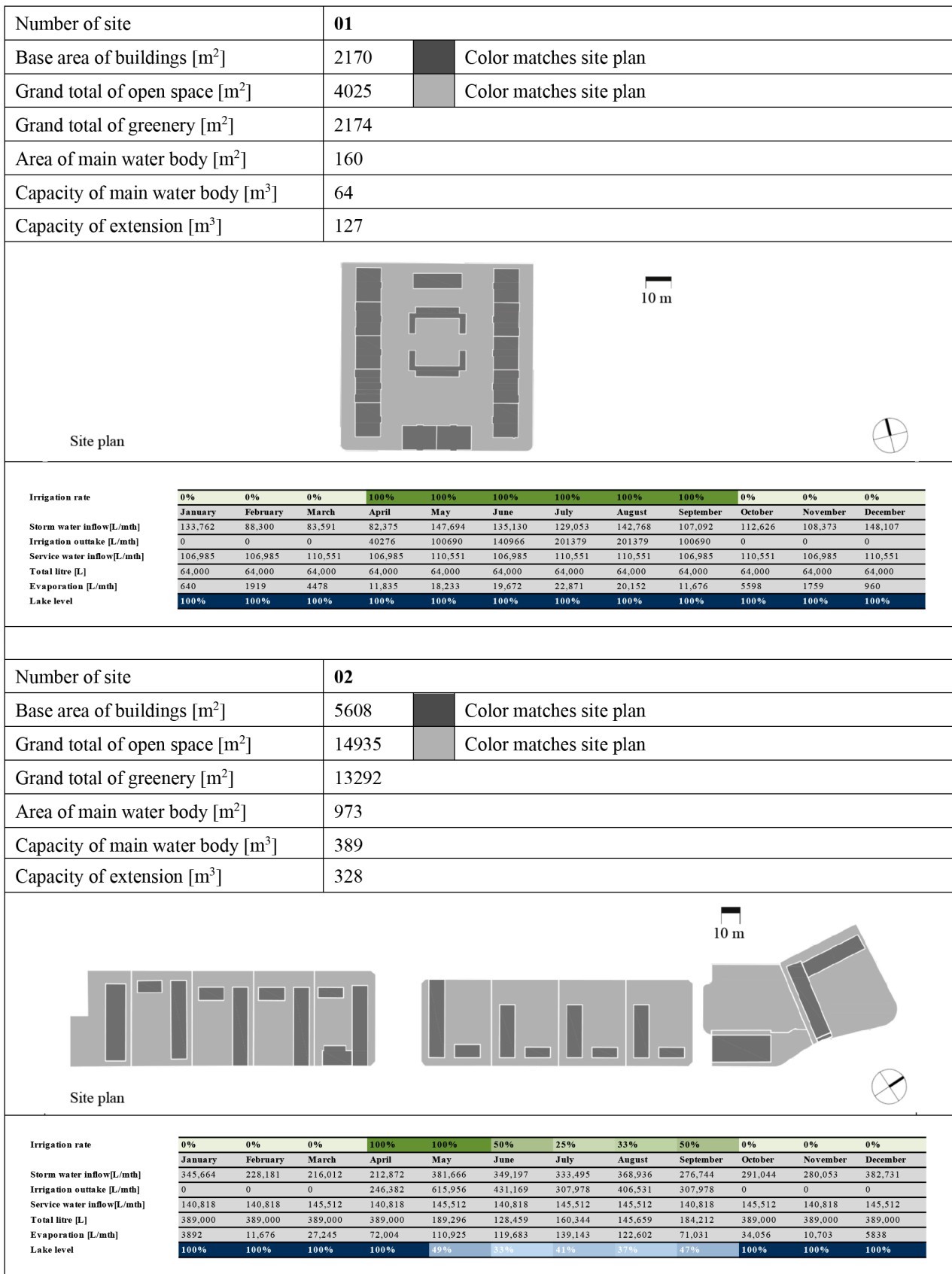

**Figure A1.** Site sheets 01 and 02.

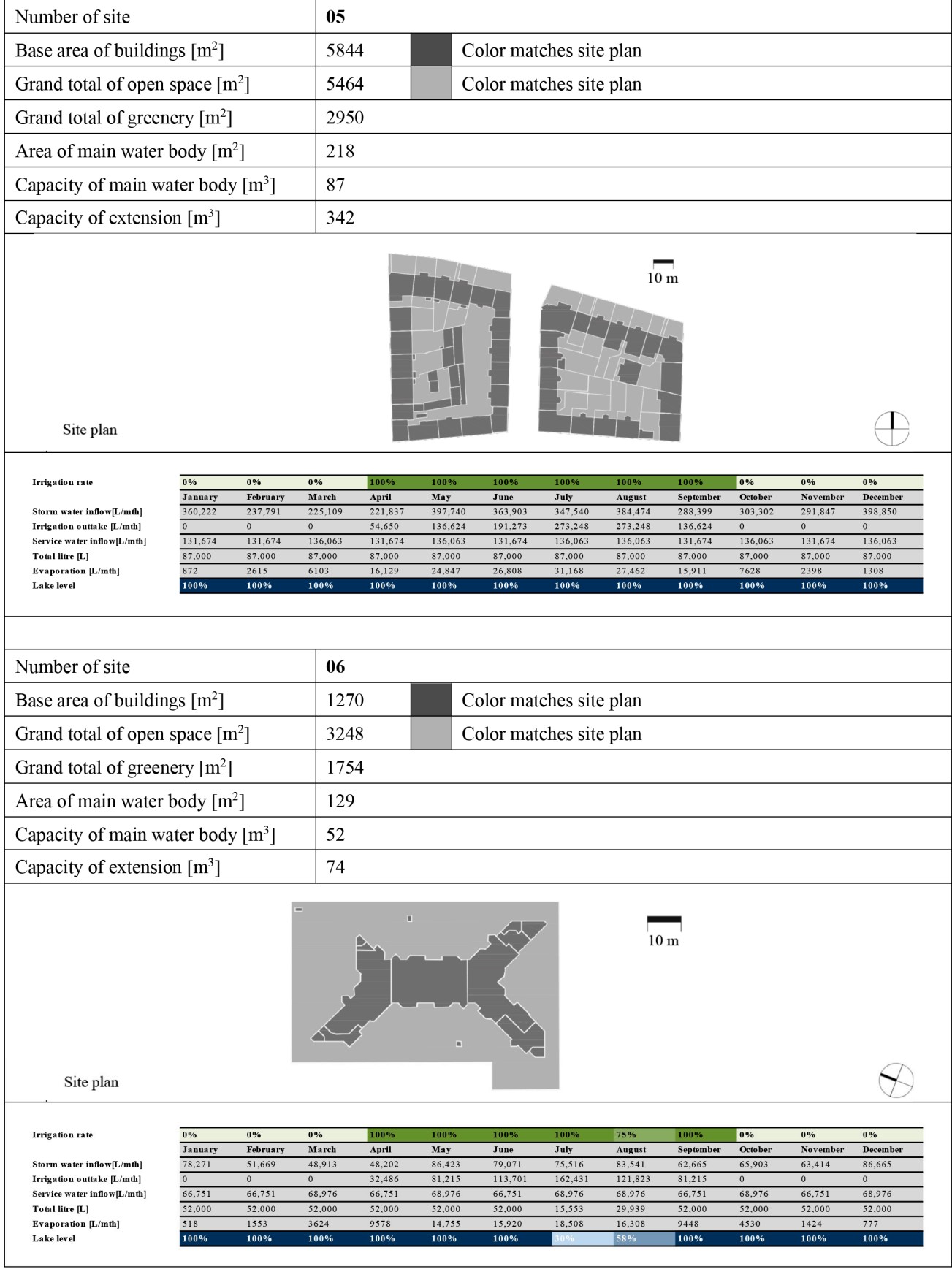

| Number of site | **05** | | |
|---|---|---|---|
| Base area of buildings [m²] | 5844 | | Color matches site plan |
| Grand total of open space [m²] | 5464 | | Color matches site plan |
| Grand total of greenery [m²] | 2950 | | |
| Area of main water body [m²] | 218 | | |
| Capacity of main water body [m³] | 87 | | |
| Capacity of extension [m³] | 342 | | |

| Irrigation rate | 0% | 0% | 0% | 100% | 100% | 100% | 100% | 100% | 100% | 0% | 0% | 0% |
|---|---|---|---|---|---|---|---|---|---|---|---|---|
| | January | February | March | April | May | June | July | August | September | October | November | December |
| Storm water inflow[L/mth] | 360,222 | 237,791 | 225,109 | 221,837 | 397,740 | 363,903 | 347,540 | 384,474 | 288,399 | 303,302 | 291,847 | 398,850 |
| Irrigation outtake [L/mth] | 0 | 0 | 0 | 54,650 | 136,624 | 191,273 | 273,248 | 273,248 | 136,624 | 0 | 0 | 0 |
| Service water inflow[L/mth] | 131,674 | 131,674 | 136,063 | 131,674 | 136,063 | 131,674 | 136,063 | 136,063 | 131,674 | 136,063 | 131,674 | 136,063 |
| Total litre [L] | 87,000 | 87,000 | 87,000 | 87,000 | 87,000 | 87,000 | 87,000 | 87,000 | 87,000 | 87,000 | 87,000 | 87,000 |
| Evaporation [L/mth] | 872 | 2615 | 6103 | 16,129 | 24,847 | 26,808 | 31,168 | 27,462 | 15,911 | 7628 | 2398 | 1308 |
| Lake level | 100% | 100% | 100% | 100% | 100% | 100% | 100% | 100% | 100% | 100% | 100% | 100% |

| Number of site | **06** | | |
|---|---|---|---|
| Base area of buildings [m²] | 1270 | | Color matches site plan |
| Grand total of open space [m²] | 3248 | | Color matches site plan |
| Grand total of greenery [m²] | 1754 | | |
| Area of main water body [m²] | 129 | | |
| Capacity of main water body [m³] | 52 | | |
| Capacity of extension [m³] | 74 | | |

| Irrigation rate | 0% | 0% | 0% | 100% | 100% | 100% | 100% | 75% | 100% | 0% | 0% | 0% |
|---|---|---|---|---|---|---|---|---|---|---|---|---|
| | January | February | March | April | May | June | July | August | September | October | November | December |
| Storm water inflow[L/mth] | 78,271 | 51,669 | 48,913 | 48,202 | 86,423 | 79,071 | 75,516 | 83,541 | 62,665 | 65,903 | 63,414 | 86,665 |
| Irrigation outtake [L/mth] | 0 | 0 | 0 | 32,486 | 81,215 | 113,701 | 162,431 | 121,823 | 81,215 | 0 | 0 | 0 |
| Service water inflow[L/mth] | 66,751 | 66,751 | 68,976 | 66,751 | 68,976 | 66,751 | 68,976 | 68,976 | 66,751 | 68,976 | 66,751 | 68,976 |
| Total litre [L] | 52,000 | 52,000 | 52,000 | 52,000 | 52,000 | 52,000 | 15,553 | 29,939 | 52,000 | 52,000 | 52,000 | 52,000 |
| Evaporation [L/mth] | 518 | 1553 | 3624 | 9578 | 14,755 | 15,920 | 18,508 | 16,308 | 9448 | 4530 | 1424 | 777 |
| Lake level | 100% | 100% | 100% | 100% | 100% | 100% | 30% | 58% | 100% | 100% | 100% | 100% |

**Figure A2.** Sites 05 and 06.

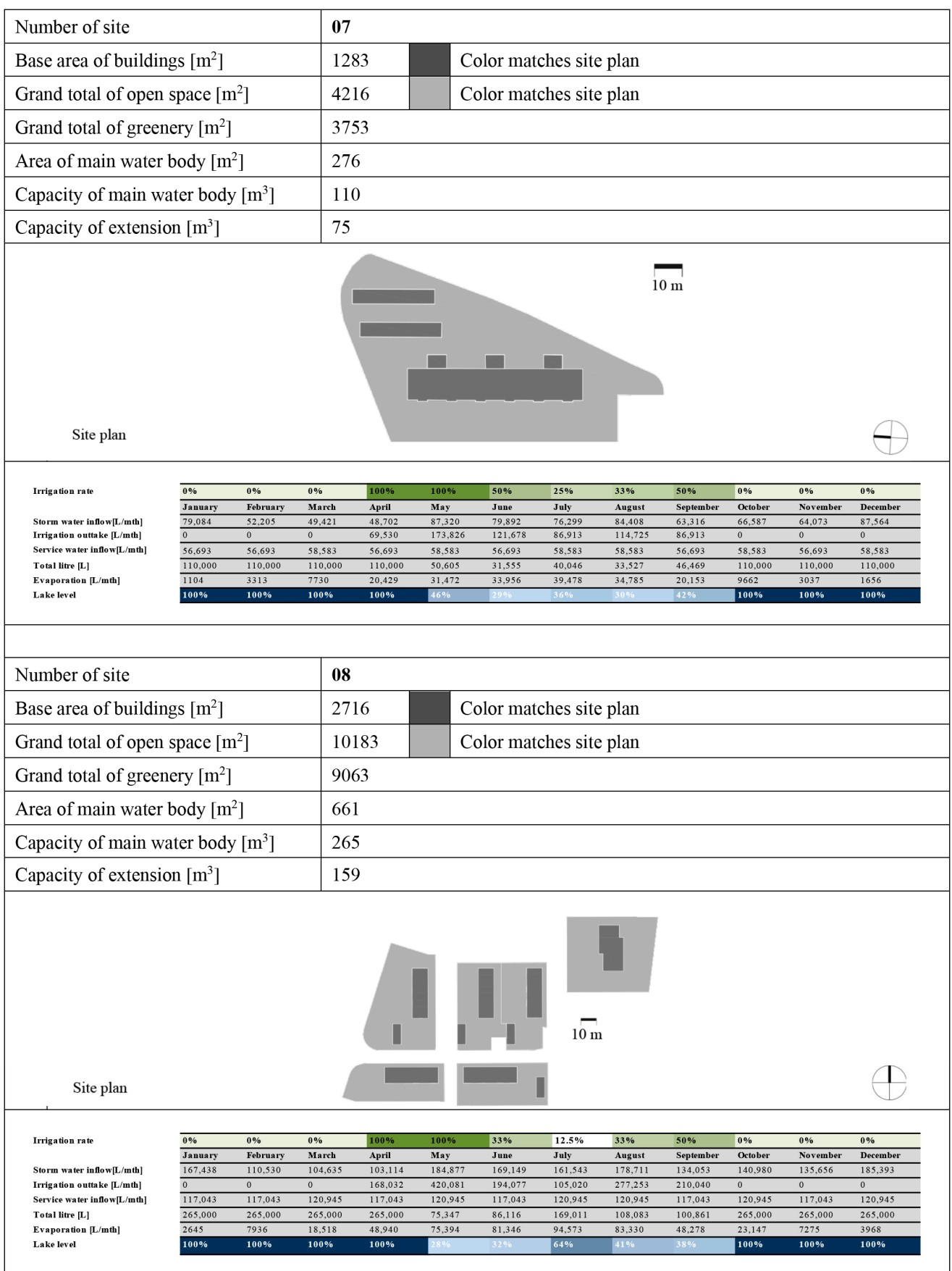

**Figure A3.** Sites 07 and 08.

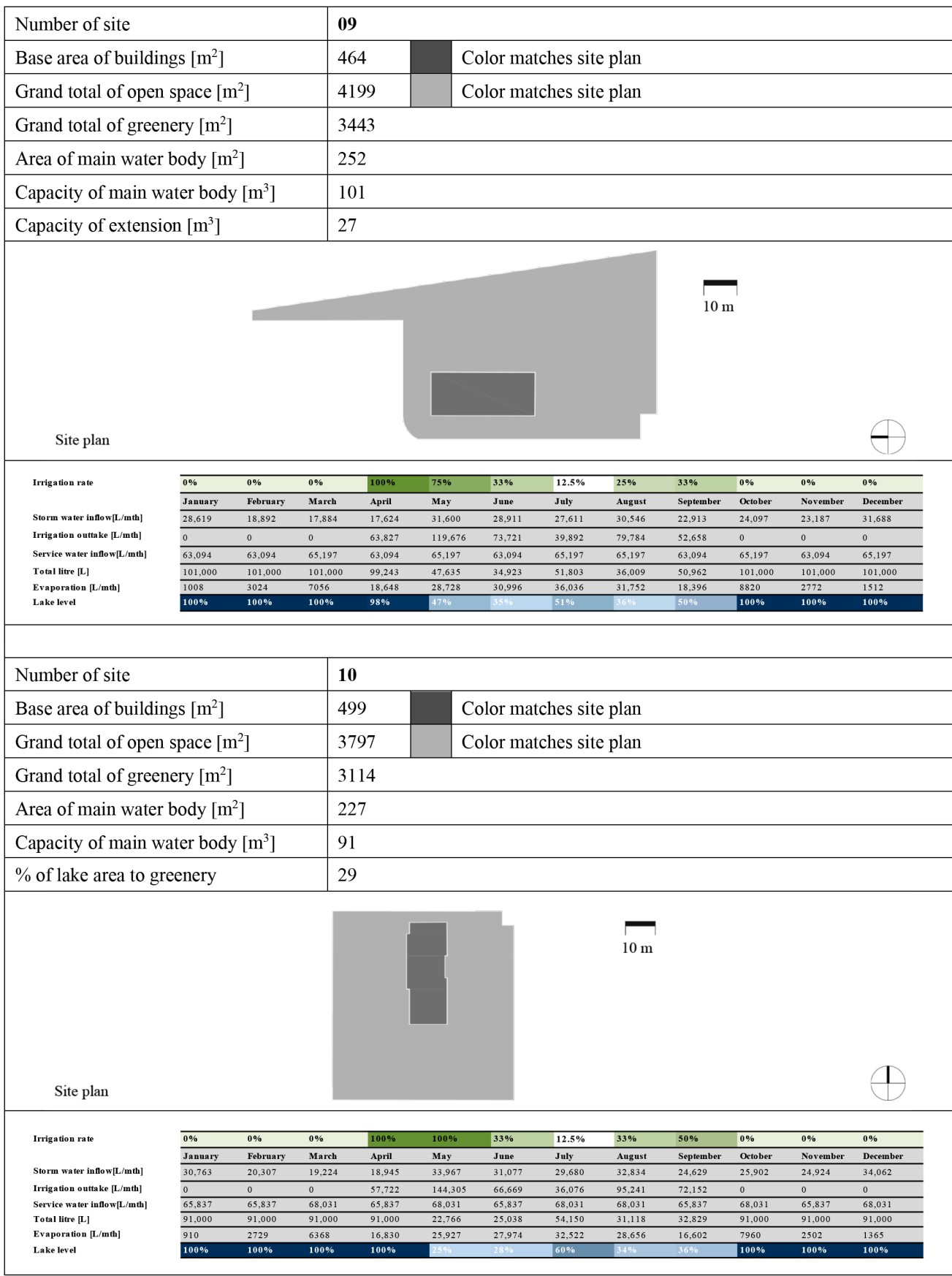

| Number of site | **09** | | |
|---|---|---|---|
| Base area of buildings [m²] | 464 | | Color matches site plan |
| Grand total of open space [m²] | 4199 | | Color matches site plan |
| Grand total of greenery [m²] | 3443 | | |
| Area of main water body [m²] | 252 | | |
| Capacity of main water body [m³] | 101 | | |
| Capacity of extension [m³] | 27 | | |

Site plan

10 m

| Irrigation rate | 0% | 0% | 0% | 100% | 75% | 33% | 12.5% | 25% | 33% | 0% | 0% | 0% |
|---|---|---|---|---|---|---|---|---|---|---|---|---|
| | January | February | March | April | May | June | July | August | September | October | November | December |
| Storm water inflow[L/mth] | 28,619 | 18,892 | 17,884 | 17,624 | 31,600 | 28,911 | 27,611 | 30,546 | 22,913 | 24,097 | 23,187 | 31,688 |
| Irrigation outtake [L/mth] | 0 | 0 | 0 | 63,827 | 119,676 | 73,721 | 39,892 | 79,784 | 52,658 | 0 | 0 | 0 |
| Service water inflow[L/mth] | 63,094 | 63,094 | 65,197 | 63,094 | 65,197 | 63,094 | 65,197 | 65,197 | 63,094 | 65,197 | 63,094 | 65,197 |
| Total litre [L] | 101,000 | 101,000 | 101,000 | 99,243 | 47,635 | 34,923 | 51,803 | 36,009 | 50,962 | 101,000 | 101,000 | 101,000 |
| Evaporation [L/mth] | 1008 | 3024 | 7056 | 18,648 | 28,728 | 30,996 | 36,036 | 31,752 | 18,396 | 8820 | 2772 | 1512 |
| Lake level | 100% | 100% | 100% | 98% | 47% | 35% | 51% | 36% | 50% | 100% | 100% | 100% |

| Number of site | **10** | | |
|---|---|---|---|
| Base area of buildings [m²] | 499 | | Color matches site plan |
| Grand total of open space [m²] | 3797 | | Color matches site plan |
| Grand total of greenery [m²] | 3114 | | |
| Area of main water body [m²] | 227 | | |
| Capacity of main water body [m³] | 91 | | |
| % of lake area to greenery | 29 | | |

Site plan

10 m

| Irrigation rate | 0% | 0% | 0% | 100% | 100% | 33% | 12.5% | 33% | 50% | 0% | 0% | 0% |
|---|---|---|---|---|---|---|---|---|---|---|---|---|
| | January | February | March | April | May | June | July | August | September | October | November | December |
| Storm water inflow[L/mth] | 30,763 | 20,307 | 19,224 | 18,945 | 33,967 | 31,077 | 29,680 | 32,834 | 24,629 | 25,902 | 24,924 | 34,062 |
| Irrigation outtake [L/mth] | 0 | 0 | 0 | 57,722 | 144,305 | 66,669 | 36,076 | 95,241 | 72,152 | 0 | 0 | 0 |
| Service water inflow[L/mth] | 65,837 | 65,837 | 68,031 | 65,837 | 68,031 | 65,837 | 68,031 | 68,031 | 65,837 | 68,031 | 65,837 | 68,031 |
| Total litre [L] | 91,000 | 91,000 | 91,000 | 91,000 | 22,766 | 25,038 | 54,150 | 31,118 | 32,829 | 91,000 | 91,000 | 91,000 |
| Evaporation [L/mth] | 910 | 2729 | 6368 | 16,830 | 25,927 | 27,974 | 32,522 | 28,656 | 16,602 | 7960 | 2502 | 1365 |
| Lake level | 100% | 100% | 100% | 100% | 25% | 28% | 60% | 34% | 36% | 100% | 100% | 100% |

**Figure A4.** Sites 09 and 10.

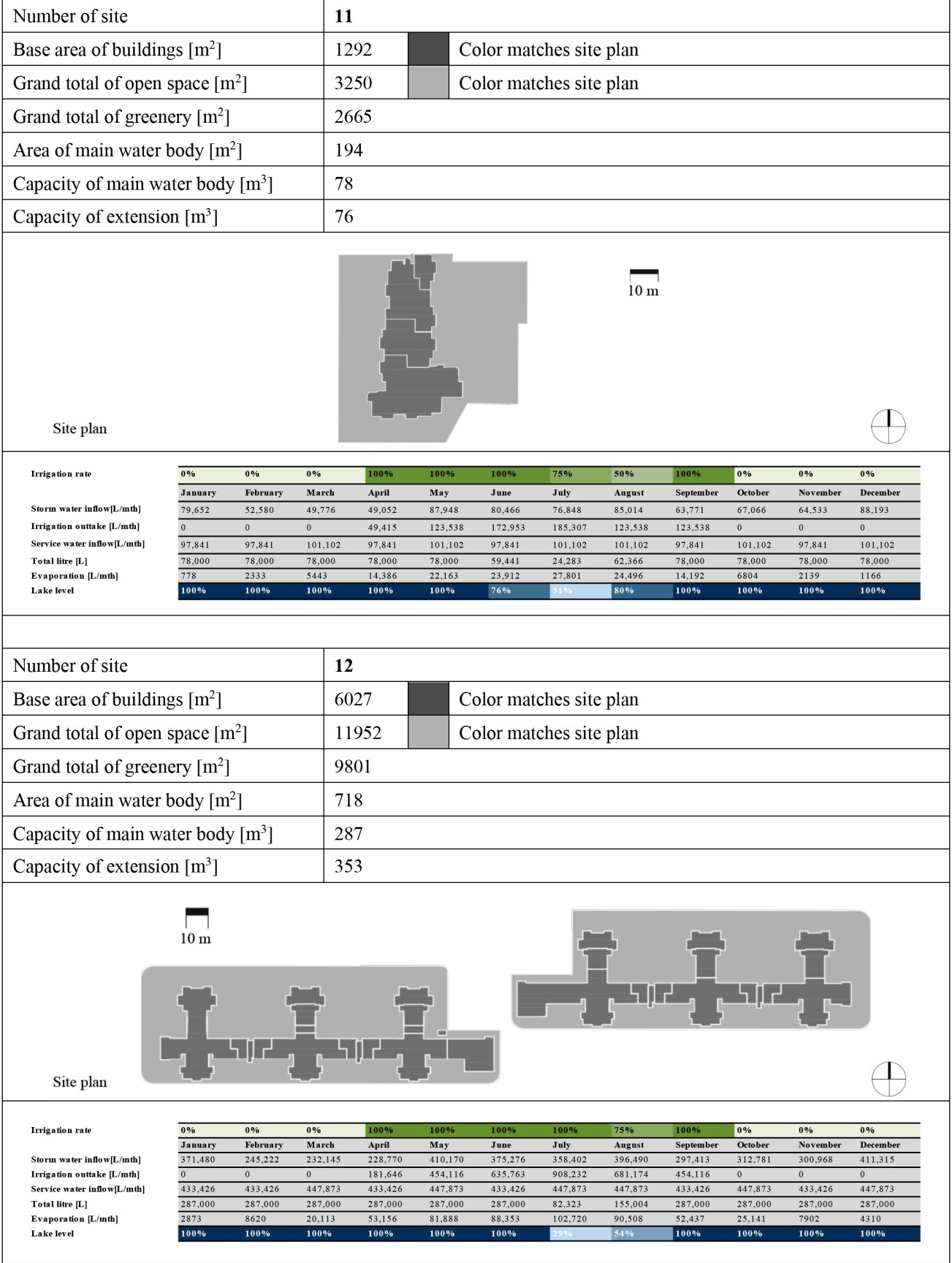

**Figure A5.** Sites 11 and 12.

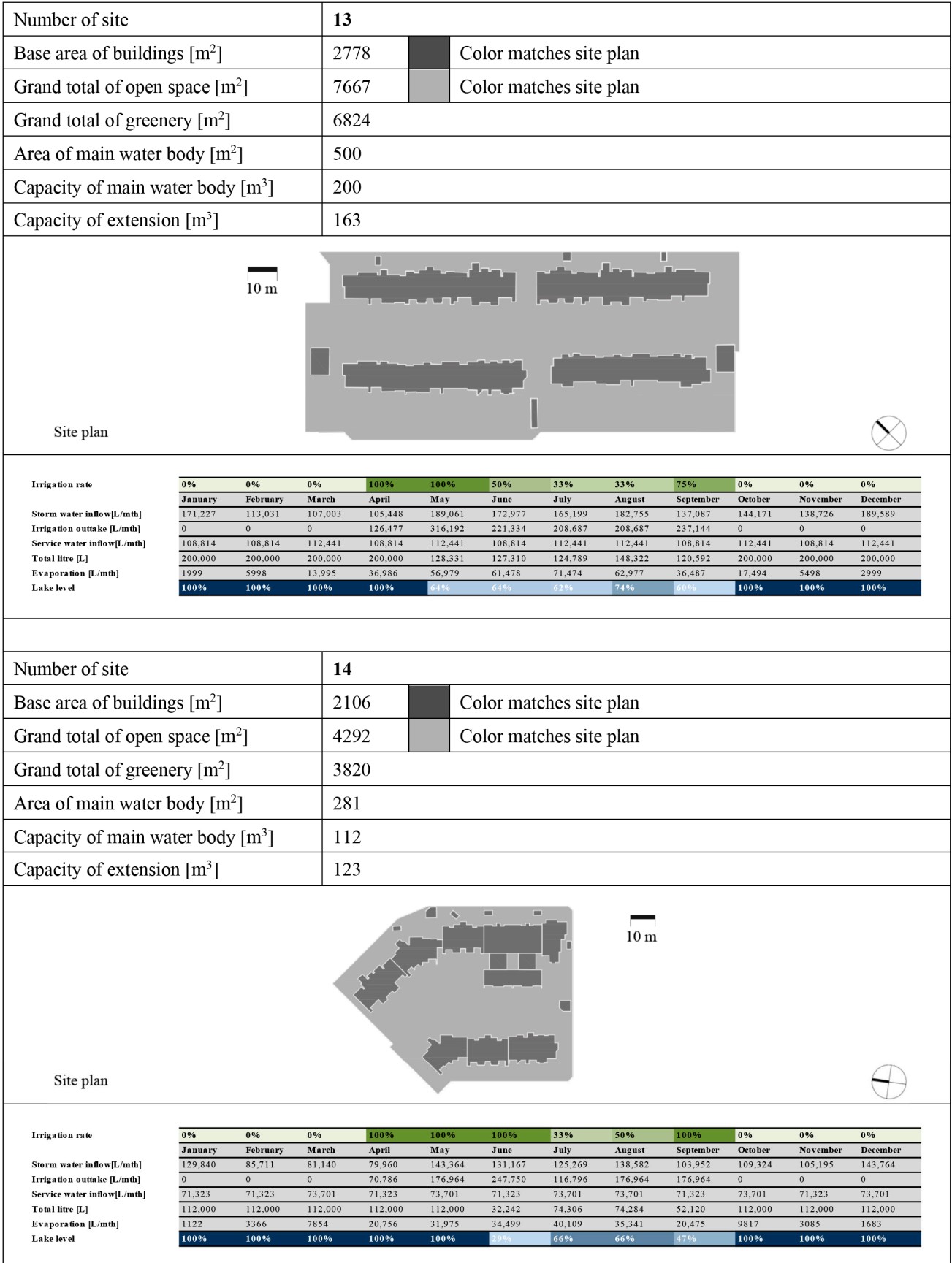

**Figure A6.** Sites 13 and 14.

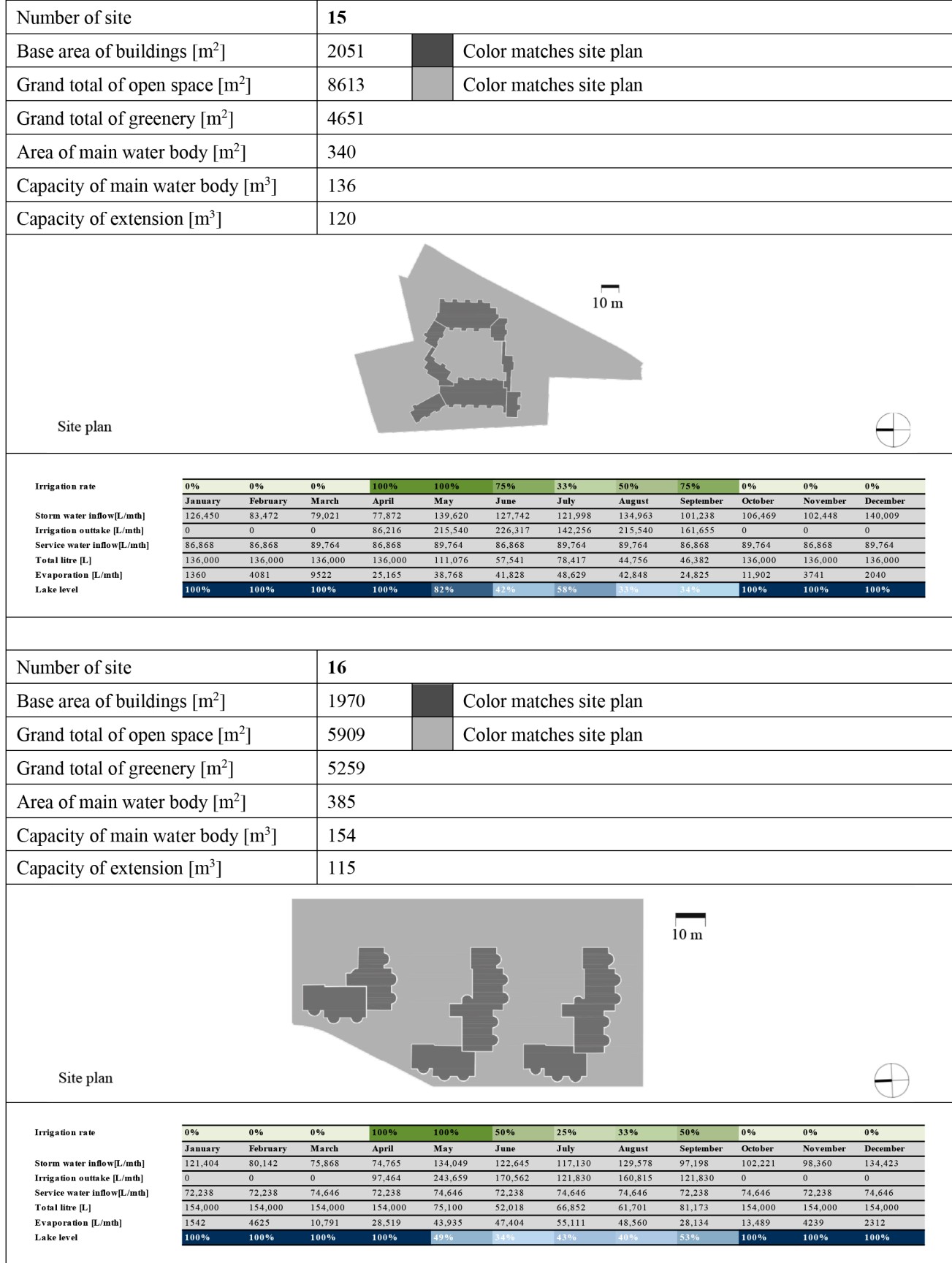

| Number of site | **15** | | |
|---|---|---|---|
| Base area of buildings [m²] | 2051 | | Color matches site plan |
| Grand total of open space [m²] | 8613 | | Color matches site plan |
| Grand total of greenery [m²] | 4651 | | |
| Area of main water body [m²] | 340 | | |
| Capacity of main water body [m³] | 136 | | |
| Capacity of extension [m³] | 120 | | |

Site plan

10 m

| Irrigation rate | 0% | 0% | 0% | 100% | 100% | 75% | 33% | 50% | 75% | 0% | 0% | 0% |
|---|---|---|---|---|---|---|---|---|---|---|---|---|
| | January | February | March | April | May | June | July | August | September | October | November | December |
| Storm water inflow[L/mth] | 126,450 | 83,472 | 79,021 | 77,872 | 139,620 | 127,742 | 121,998 | 134,963 | 101,238 | 106,469 | 102,448 | 140,009 |
| Irrigation outtake [L/mth] | 0 | 0 | 0 | 86,216 | 215,540 | 226,317 | 142,256 | 215,540 | 161,655 | 0 | 0 | 0 |
| Service water inflow[L/mth] | 86,868 | 86,868 | 89,764 | 86,868 | 89,764 | 86,868 | 89,764 | 89,764 | 86,868 | 89,764 | 86,868 | 89,764 |
| Total litre [L] | 136,000 | 136,000 | 136,000 | 136,000 | 111,076 | 57,541 | 78,417 | 44,756 | 46,382 | 136,000 | 136,000 | 136,000 |
| Evaporation [L/mth] | 1360 | 4081 | 9522 | 25,165 | 38,768 | 41,828 | 48,629 | 42,848 | 24,825 | 11,902 | 3741 | 2040 |
| Lake level | 100% | 100% | 100% | 100% | 82% | 42% | 58% | 33% | 34% | 100% | 100% | 100% |

| Number of site | **16** | | |
|---|---|---|---|
| Base area of buildings [m²] | 1970 | | Color matches site plan |
| Grand total of open space [m²] | 5909 | | Color matches site plan |
| Grand total of greenery [m²] | 5259 | | |
| Area of main water body [m²] | 385 | | |
| Capacity of main water body [m³] | 154 | | |
| Capacity of extension [m³] | 115 | | |

Site plan

10 m

| Irrigation rate | 0% | 0% | 0% | 100% | 100% | 50% | 25% | 33% | 50% | 0% | 0% | 0% |
|---|---|---|---|---|---|---|---|---|---|---|---|---|
| | January | February | March | April | May | June | July | August | September | October | November | December |
| Storm water inflow[L/mth] | 121,404 | 80,142 | 75,868 | 74,765 | 134,049 | 122,645 | 117,130 | 129,578 | 97,198 | 102,221 | 98,360 | 134,423 |
| Irrigation outtake [L/mth] | 0 | 0 | 0 | 97,464 | 243,659 | 170,562 | 121,830 | 160,815 | 121,830 | 0 | 0 | 0 |
| Service water inflow[L/mth] | 72,238 | 72,238 | 74,646 | 72,238 | 74,646 | 72,238 | 74,646 | 74,646 | 72,238 | 74,646 | 72,238 | 74,646 |
| Total litre [L] | 154,000 | 154,000 | 154,000 | 154,000 | 75,100 | 52,018 | 66,852 | 61,701 | 81,173 | 154,000 | 154,000 | 154,000 |
| Evaporation [L/mth] | 1542 | 4625 | 10,791 | 28,519 | 43,935 | 47,404 | 55,111 | 48,560 | 28,134 | 13,489 | 4239 | 2312 |
| Lake level | 100% | 100% | 100% | 100% | 49% | 34% | 43% | 40% | 53% | 100% | 100% | 100% |

**Figure A7.** Sites 15 and 16.

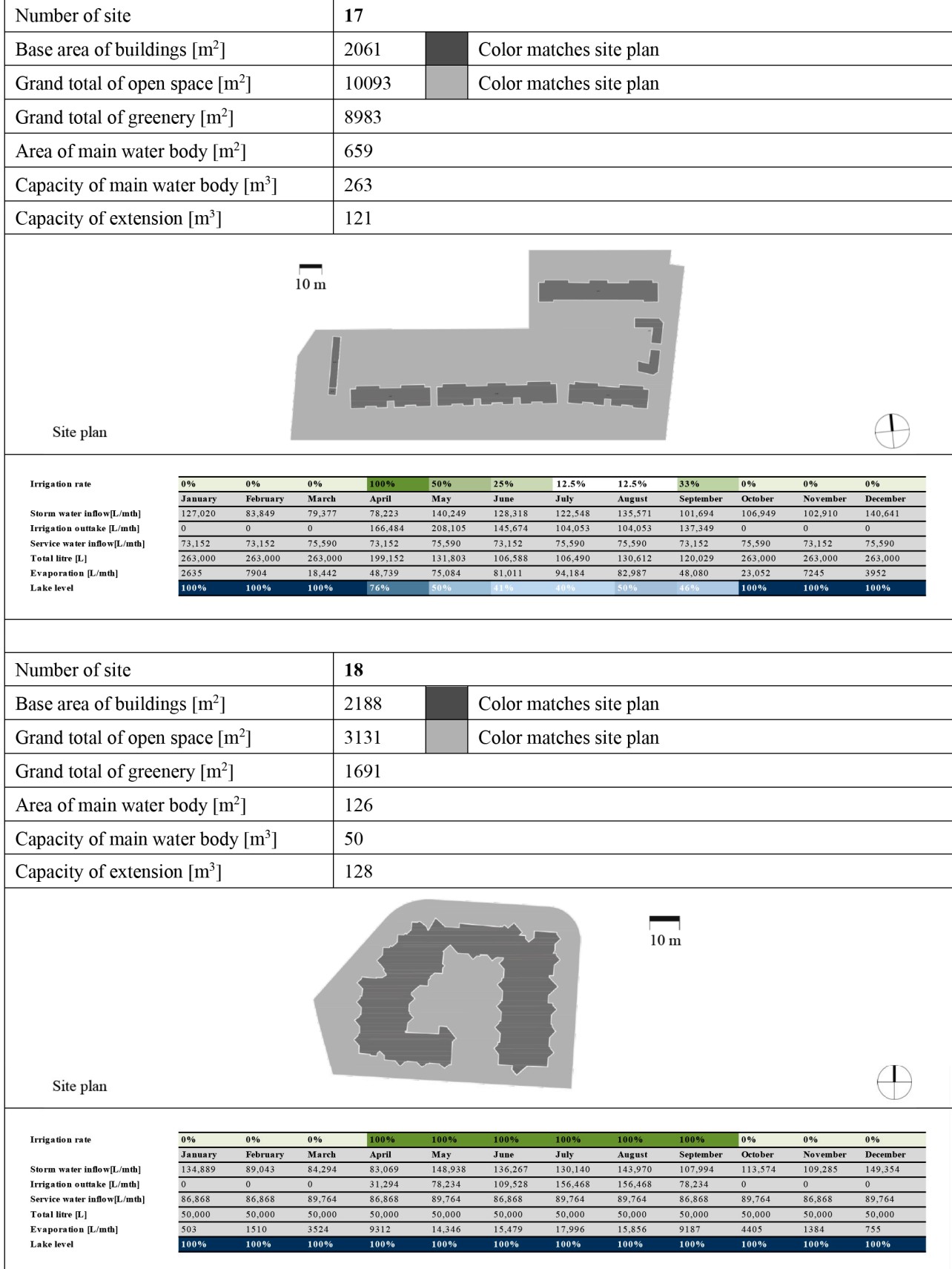

**Figure A8.** Sites 17 and 18.

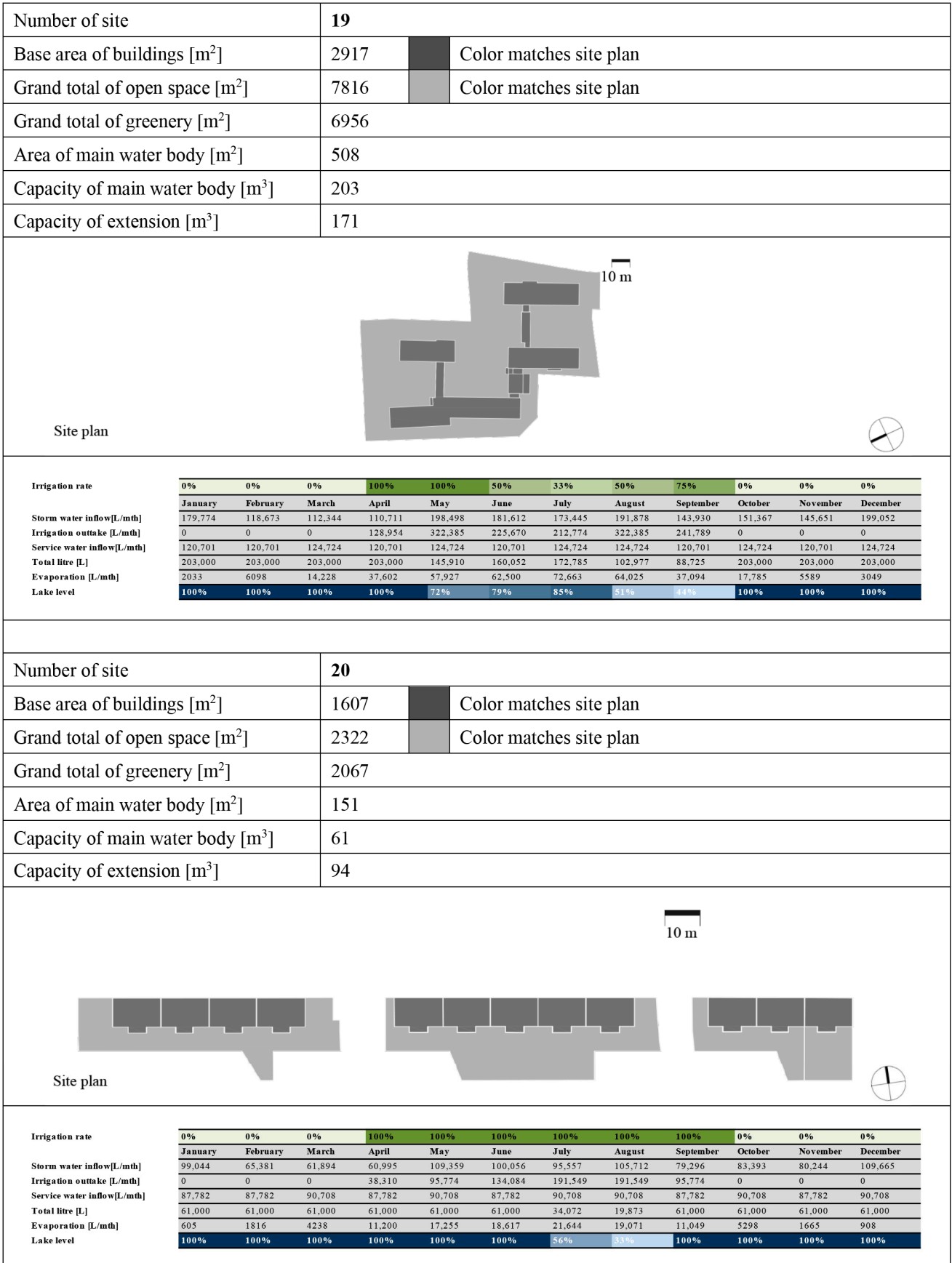

**Figure A9.** Sites 19 and 20.

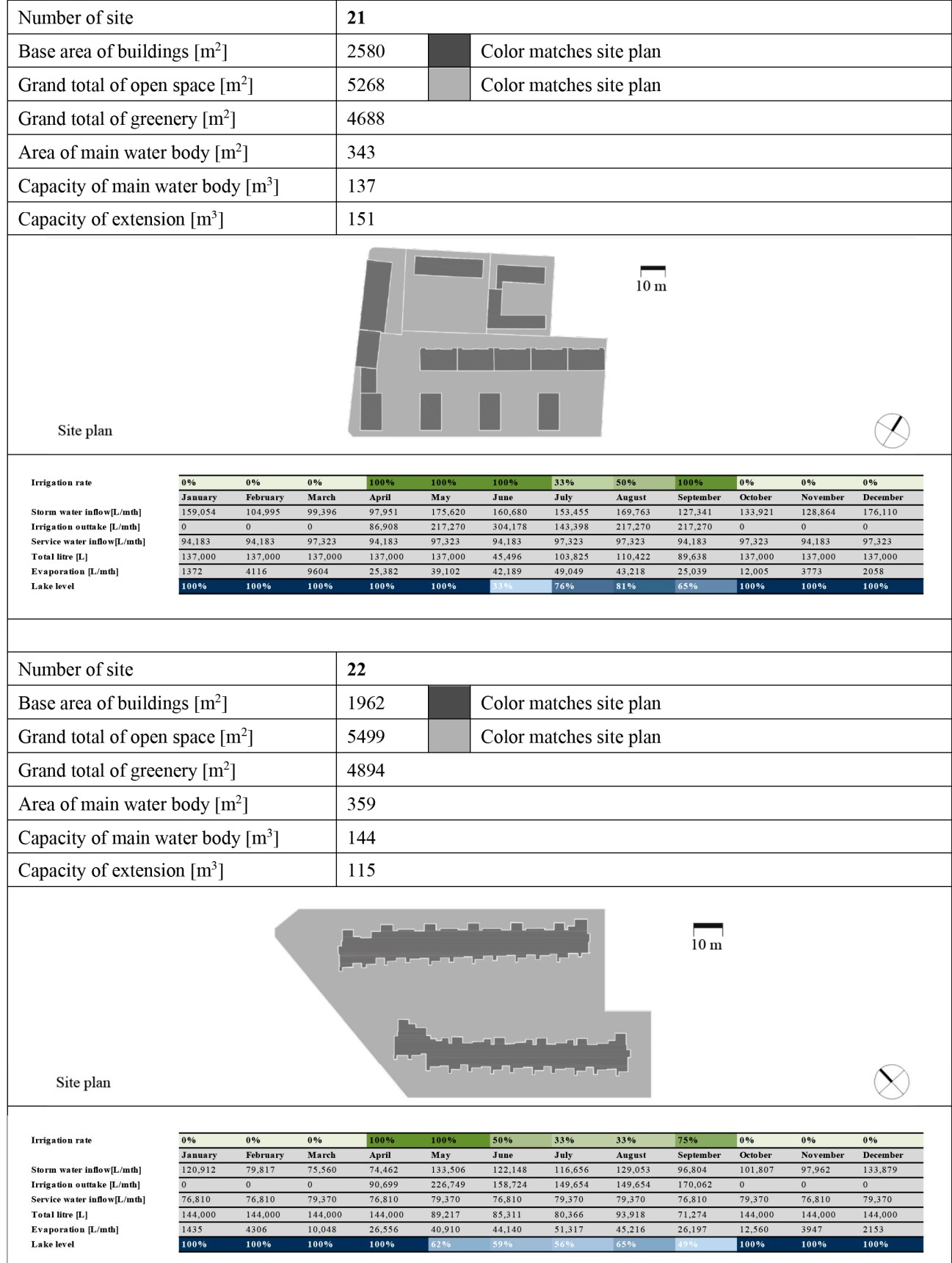

**Figure A10.** Sites 21 and 22.

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
