# Peer review of "From District to City Scale: The Potential of Water-Sensitive Urban Design (WSUD)"

_water, doi:10.3390/w16040582_

Round 1
Reviewer 1 Report
Comments and Suggestions for Authors
This research paper titled "From district to city scale: the potential of Water Sensitive Urban Design (WSUD)" explores the concept of WSUD as a means to address the challenges posed by climate change in urban environments. The paper presents a prototype of WSUD implemented in the city of Darmstadt and investigates its suitability, potential impact on the city's water cycle, and relevance for climate change adaptation. Overall, the paper provides valuable insights into the application of WSUD and its potential benefits. However, there are some areas that could be improved to enhance the clarity and comprehensiveness of the research.
Research Gap and Contribution:
The paper lacks a clear identification of the research gap and how this study contributes to filling that gap. It would be helpful to explicitly state the specific knowledge or practical gap in the field of WSUD or urban water management that this research aims to address. Additionally, highlighting the unique aspects or innovative elements of the proposed WSUD prototype in Darmstadt would strengthen the paper's contribution to the existing literature.
Methodological Rigor:
The paper would benefit from a more detailed description of the methodology used to derive the site suitability indicators and calculate the potential impact of the WSUD prototype. This includes providing a comprehensive explanation of the data sources, criteria used for site selection, and the modeling or analytical techniques employed. Without this information, it is difficult to assess the robustness and reliability of the findings. Additionally, sensitivity analysis or validation of the results against alternative scenarios or models would add further credibility to the study.
Quantitative Analysis and Interpretation:
The paper provides quantitative results, such as the number of suitable sites and potential water savings, but lacks in-depth analysis and interpretation of these findings. It is important to discuss the implications of the results in relation to the objectives of WSUD, urban water management, and climate change adaptation. For example, how do the identified sites contribute to overall water resilience in the city? What are the potential challenges or trade-offs associated with implementing WSUD at a larger scale? Providing a more nuanced analysis and interpretation of the results will strengthen the paper's contribution and practical relevance.
Discussion of Limitations:
The paper does not adequately address the limitations of the study. It is crucial to acknowledge the inherent limitations and potential sources of bias or uncertainty in the methodology, data, and assumptions used. This will provide a more balanced perspective and allow readers to assess the reliability and generalizability of the findings. Additionally, discussing any potential constraints or barriers to the implementation of WSUD in real-world scenarios would contribute to a more comprehensive discussion.
Policy and Practical Implications:
The paper briefly mentions the potential benefits of WSUD, such as water savings, improved livability, and climate resilience. However, it would be valuable to discuss the broader policy and practical implications of implementing WSUD at the city scale. How does WSUD align with existing urban planning policies and strategies? What are the potential economic, social, and environmental implications of large-scale WSUD implementation? Addressing these aspects will enhance the practical relevance and applicability of the research.
Future Research Directions:
The paper could conclude with a section highlighting potential avenues for future research. This can include areas that were not addressed in the current study, unresolved questions or challenges, and suggestions for further improving the WSUD concept or its implementation. This will encourage further exploration and development of the topic.
By addressing these critical comments and incorporating the suggested improvements, the paper will become more robust, comprehensive, and impactful in advancing the field of Water Sensitive
Comments on the Quality of English LanguageWhile the overall language and writing style are clear, there are instances where the paper uses technical jargon or abbreviations without sufficient explanation. It is important to ensure that the paper is accessible to a broad readership, including those outside the specific field of WSUD. Providing clear definitions and explanations of key terms and abbreviations throughout the text will improve the readability and comprehension of the paper.
Reviewer 2 Report
Comments and Suggestions for Authors
Introduction: The paper presents a comprehensive overview of the severe impacts of the hottest and driest summer of 2022 in Germany, emphasizing the importance of addressing climate change-driven extreme weather events. The focus is on the water crisis, affecting not only nature and the economy but also public health. The introduction needs effective discussion on the need for proactive measures, particularly in densely populated urban areas.
Please consider the following relative references:
https://doi.org/10.1016/j.jhydrol.2016.01.007
https://doi.org/10.2166/wp.2017.182
Water Sensitive Urban Design (WSUD) Concept: The paper introduces the Water Sensitive Urban Design (WSUD) as a potential solution to mitigate the challenges posed by climate change, with a specific emphasis on heat-proofing and water-securing urban developments. The incorporation of an artificial lake as part of an Integrated Water Resource Management System (IWRMS) is a key highlight. The use of treated grey water and stormwater from housing blocks demonstrates a practical application of WSUD principles but needs further support from references.
Prototype Development: The paper builds upon a prototype of WSUD centered around an artificial lake, providing a detailed description of its components, such as water reclamation, retention, treatment, and distribution. The integration of the system with housing blocks and its reliance on treated grey water and stormwater showcase a holistic approach to sustainable urban development. The prototype serves as a model for potential replication in other urban areas. Transferability limitations need discussion.
Site Suitability Analysis: The incorporation of indicators for site suitability assessment adds a practical dimension to the paper. The application of these indicators to identify potential locations for replicable WSUD projects in the city of Darmstadt is a commendable effort. The results, indicating 22 suitable sites for prototype implementation, offer a tangible outcome that policymakers and urban planners can use for informed decision-making. Any water resource balances would be appreciated by the readers.
Benefits of WSUD Implementation: The paper effectively communicates the multifaceted benefits of WSUD implementation. These include substantial water savings, stormwater retention, adiabatic cooling during heatwaves, increased biodiversity, and improved overall livability of the sites and the city. Quantifying the potential water savings (147 million liters) and irrigation water provision (24 million liters) enhances the paper's practical relevance. Discuss calculations uncertainty using statistical analysis.
Conclusion: The paper concludes with a strong emphasis on the positive impact of WSUD in addressing climate change-related challenges in urban environments. The findings from the prototype and site suitability analysis in Darmstadt provide valuable insights for policymakers, urban planners, and researchers working towards sustainable urban development. Overall, the paper effectively combines theoretical concepts with practical applications, contributing to the discourse on climate change adaptation in urban settings.
Round 2
Reviewer 1 Report
Comments and Suggestions for Authors
The paper has made significant improvements, but further integration of references and discussions around urban water security and integrated urban water management would enhance its contribution to the field. Ensure that the suggested literature is appropriately cited throughout the paper to reinforce the theoretical foundations of the WSUD prototype.
The introduction provides a comprehensive overview of the challenges posed by extreme weather events and the need for water-sensitive urban design. However, it would greatly enhance the paper to explicitly link water-sensitive design to broader concepts such as urban water security and integrated urban water management. Suggest incorporating references, particularly from the following works:
https://www.nature.com/articles/s41467-021-25026-3 https://www.sciencedirect.com/science/article/pii/S221146452200015X https://www.sciencedirect.com/science/article/pii/S1364815216310623 https://www.mdpi.com/2079-9276/8/4/178
minor
Author Response
"Please see the attachment"

Reviewer 2 Report
Comments and Suggestions for Authors
THE AUTHORS IMPROVED THE PAPER
Author Response
"Please see the attachment".
